# Heavy-boundary mode patterning and dynamics of topological phonons in polymer chains and supramolecular lattices on surfaces

José D. Cojal González [1], Jakub Rondomanski[2], Konrad Polthier[2], Jürgen P. Rabe [1] & Carlos-Andres Palma [1,3] ✉

In topological band theory, phonon boundary modes consequence of a topologically non-trivial band structure feature desirable properties for atomically-precise technologies, such as robustness against defects, wave-guiding, and one-way transport. These topological phonon boundary modes remain to be studied both theoretically and experimentally in synthetic materials, such as polymers and supramolecular assemblies at the atomistic level under thermal fluctuations. Here we show by means of molecular simulations, that surface-confined Su-Schrieffer-Heeger (SSH) phonon analogue models express robust topological phonon boundary modes at heavy boundaries and under thermal fluctuations. The resulting bulk-heavy boundary correspondence enables patterning of boundary modes in polymer chains and weakly-interacting supramolecular lattices. Moreover, we show that upon excitation of a single molecule, propagation along heavy-boundary modes differs from free boundary modes. Our work is an entry to topological vibrations in supramolecular systems, and may find applications in the patterning of phonon circuits and realization of Hall effect phonon analogues at the molecular scale.

The study of topology in the context of electronic band theory and corresponding topological phases of matter is widespread[1–8] Equivalent topological concepts have been also applied to phononic systems[9–18], in both crystalline materials[19–24] and periodic artificial structures[25–33], with the aim of identifying distinct phenomena in phononic phases of matter which can be classified by so-called topological indexes or invariants. Yet vibrational and related dynamic and topological phonon phases, especially at finite temperatures and in the presence of disorder, remain to be explored in atomistic soft matter i.e. molecule-based materials and synthetic materials such as polymers, self-assembled networks, metal-organic frameworks and covalent organic frameworks, to mention a few. Phonon bands are a universal property of many extended molecule-based materials[34,35], a highly diverse set of materials which can be built from millions of compounds[36]. Thus, the prospect of chemical compounds belonging to two different sets of phonon phases of matter−a topological phase and non-topological phase−could bear far-reaching implications.

Topology in physics and chemical physics, distinguishes itself from topology in chemistry, which rather focuses on the study of shapes in three-dimensional (3D) as well as in connectivity space[35,37–43]. In physics, topology deals mainly with the mathematical study of the equations of motion and eigenfunctions, usually in the context of classifications of matrix operators and band theory[44–46] leading to various phases depending on their classification. For a quasi-1D

[1]Department of Physics & IRIS Adlershof, Humboldt-Universität zu Berlin, Berlin, Germany. [2]Department of Mathematics and Computer Science, Freie Universität Berlin, Berlin, Germany. [3]Institute of Physics, Chinese Academy of Sciences, Beijing, P. R. China. ✉e-mail: palma@iphy.ac.cn

material, like a linear polymer chain for instance, diverse topological classification frameworks can be employed to define topological invariants, such as holonomy groups, winding numbers and geometric (Berry) phases[35,47,48]. One topological classification in differential geometry entails classification according to the holonomy group elements of the vibrational space of coupled oscillators[48]. A more common classification relies on algebraic topology, whereby the equations of motion for coupled oscillators are expressed in an ordinary differential equation akin to the Schrödinger eigenproblem, and classified by algebraic symmetries. This approach establishes a correspondence between phonons (bosons) and electrons (fermions) band topology[49,50]. Other strategies rely on identifying non-trivial topological invariants in opposition to the 'atomic limit'[46], wherein atoms are disconnected. The latter approach has been employed to search for non-trivial phonon topology in crystallographic databases[51–53].

Distinct phenomena can emerge in topological phases, such as boundary modes with intriguing properties, expressing for example robustness against local defects, nonreciprocity or unidirectionality of vibrational excitations[12,14,49,54–57]. Thus far, mostly mechanical models, such as Maxwell lattices[12,58], have been employed to engineer and demonstrate such vibrational topological properties[59,60]. These toy-models consisting of masses, springs, bars and plates offer a minimal framework for the design of metamaterials with applications in acoustics[25,28,61,62], robotics[63], thermal diodes[50,64] and waveguiding[65–67], among others. Despite recent experiments on topological phonon modes in graphene[68] and some transition metal monosilicides[69,70], topological phonon boundary modes at the atomic-scale have not been demonstrated, and approaches towards their engineering akin to metamaterials remain elusive. Topological vibrational boundary modes (TBM) related to the Su-Schrieffer-Heeger (SSH) model[71] are well-known in mechanical systems consisting of spring chains[12,72]. The identification of SSH-like and additional topological phonon phases in molecule-based synthetic materials could serve as a departure point to realise functional topological phononics at the atomic-scale, with potential in the engineering of polymerisation dynamics[9], thermal management[64], superconductivity enhancement[73], negative spring constant design[74] and phonon circuitry to mention a few. However, extended molecule-based materials are usually described by unit cells of hundreds of atoms, rendering topological classification and determination of boundary modes challenging. Moreover, in order to predict topological phonon phases in soft matter, at least four steps are needed: revision or development of mathematical topological models which capture key symmetries and dimensions, addressing the effect of disorder and temperature on said models, followed by assessment of electronic and finally quantum corrections. Yet for polymers alone, mathematical models are extensive[75]. Therefore, surface-confinement e.g. from on-surface self-assembly fabrication strategies, offers means to define symmetries and reduce dimensionality of complex matter[76,77]. The versatility of supramolecular interactions in self-assembly[78] can be further employed to tailor effective spring constants of vibrational matter, thereby leveraging topological design in semiconducting[79,80], sensing[81] or switching[82–84] applications of precision supramolecular lattices[85–87].

Here, we introduce surface-confined SSH dynamical matrix point-mass models and equivalent atomistic molecular dynamics (MD) simulations, to explore topological phases under thermal fluctuations in polymers and supramolecular self-assembled lattices. We demonstrate that topological phonon phases host topological phonons at heavy boundaries in polymers and axial coordination chains, paving the way for realising experimental platforms which may decouple vibrations from surfaces. Specifically, our work is presented in four main sections. First, we recall the SSH phonon analogue (pSSH) for the study of topological vibrational modes in polymers and supramolecular lattices. We elaborate the correspondence between the pSSH topology and its topological heavy-

boundary mode and explore the effect of electronic structure by studying realistic topological phonon phases in polyynes by density-functional tight binding (DFTB) methods. Second, we introduce the adsorbed SSH (aSSH) model on an implicit surface and corresponding topologically non-trivial boundary modes in atomistic simulations under thermal fluctuations. We provide the trivial case counter-example in Supplementary Information. Third, we describe non-trivial boundary modes of a double-chain adsorbed SSH (daSSH) model, and the counter examples in the Supplementary Information. Finally, we discuss how to identify topological phonon phases of the daSSH model by patterning topological heavy-boundary modes in all-atomistic axial coordination supramolecular chains, and study the effect of topological mode excitation and propagation by comparison to a free boundary mode on such arrays.

Our results reveal a topological boundary patterning principle through the bulk-heavy boundary correspondence, phonon topology under thermalised conditions and establishes supramolecular systems as a functional platform for designing and patterning topological physics, bridging the gap between vibrations in organic chemistry and condensed matter physics.

## Results

### Phonon band topology and respective bulk-heavy boundary correspondence in the 1D SSH phonon analogue

Numerous chemical compounds can be accurately represented by spring-mass mechanical models, which allow for the expression of topological vibrational modes. The exemplary case is a linear spring chain consisting of alternating strong (stiff) and weak (soft) springs[12,49,57,63,72,88], the so-called pSSH model (Fig. 1a, b). In the following, we extend the pSSH model to adsorbed polymers at finite temperatures (Fig. 1c–f) and introduce the heavy-boundary mode patterning principle (Fig. 1g). These models and simulations aim to design vibrational states unique to the sequence of strong and weak spring constants in a material, as opposed to vibrational states pertaining to a local end group. Topological vibrational modes may originate due to eigenvalue (band) inversion[26,89], whereby a low-energy vibrational mode of the material moves to a high-energy mode via an eigenvalue crossing (Supplementary Fig. 2) as characterised by a winding number of 1 (Supplementary Fig. 3). In the pSSH model, a boundary created by a heavy mass (M) connected by a stiff spring, hosts a TBM, not only because of the stiff spring itself, but due to an alternating sequence of stiff and soft springs in the polymer or lattice. Conversely, if the polymer is connected to the heavy boundary (M) by a soft spring, there are no TBMs (Fig. 1b, d, f). Boundary modes are typically zero-temperature eigenvalue problems solved by means of, e.g., dynamical matrices and normal mode analysis (NMA). Atomistic classical molecular mechanics and related dynamic approaches can be further employed to study the robustness of topological boundary modes under thermal fluctuations. Specifically, we use the dynamical matrix approach[90] to calculate band spectra for point-mass models and NMA for the atomistic eigenmode problem in minimum energy configurations (see Methods). Moreover, the Power Spectral Density (PSD) bands are calculated using the autocorrelation function of the velocities obtained from MD simulations, thus obtaining the phonon band structure for each simulation at finite temperature. In this way, we can relate the phonon spectra of the thermalised models with the band structure of the static configuration (see Methods section). Supplementary Fig. 4 depicts the finite-temperature phonon band of the pSSH heavy-boundary model. It follows well-known studies on the SSH model[49], with the difference that we demonstrate that the bulk-boundary correspondence (indicator of a topological boundary mode) holds when creating a heavy boundary (below). Similar studies have shown the bulk-boundary correspondence of the phonon SSH model by employing a fixed boundary[72] and without temperature fluctuations.

$\kappa_1 > \kappa_2$ Topological vibrational boundary modes in polymers and supramolecular lattices

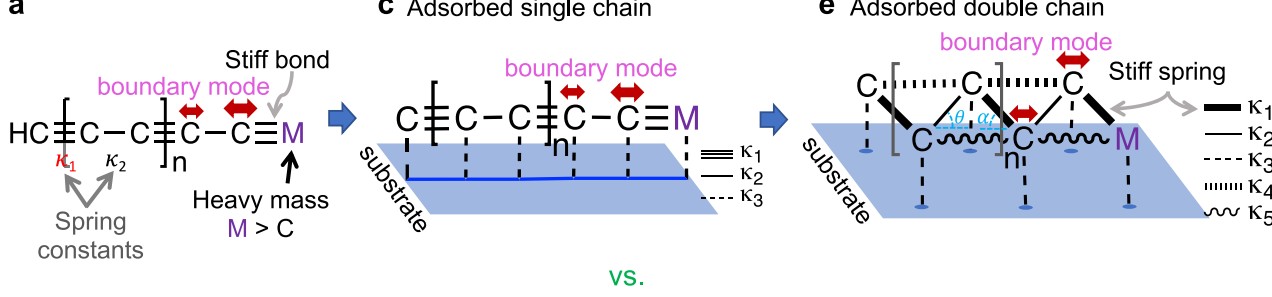

vs.

$\kappa_1 < \kappa_2$ Trivial isomers without boundary modes (Supplementary Information)

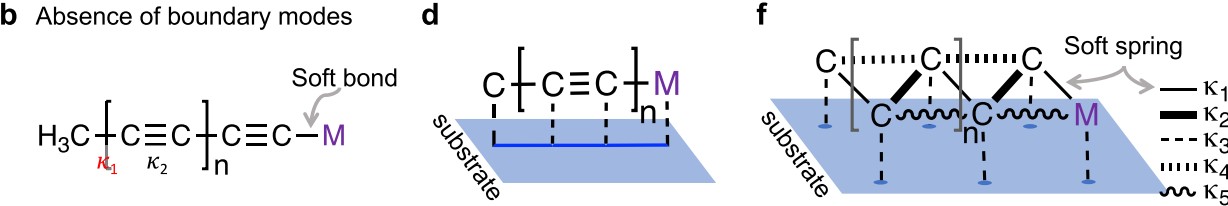

**g** Heavy-boundary patterning of topological modes

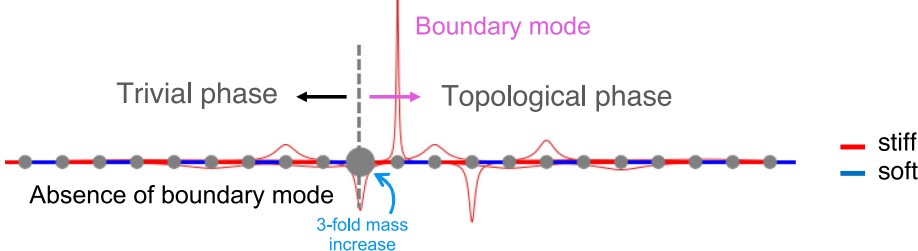

**Fig. 1 | Topological phases, their vibrational boundary modes and heavy-boundary mode patterning in phonon Su-Schrieffer-Heeger (SSH) analogue models. a** A topological vibrational boundary mode occurs when a heavy boundary is formed in polymers or lattices of coupled chains, with alternating weak/strong (or soft/stiff) spring constants (denoted by $\kappa$), terminating in a stiff spring connected to a heavier atom (M) (a topological SSH phase); as opposed to **b**. a soft spring termination. These principles can be extended to **c, d**. a chain adsorbed on a surface (aSSH model), and **e, f** a double chain (daSSH model). **g** A heavy-boundary can be employed to pattern a topological mode at a desired point in a polymer chain . The exponentially localised vibrations spanning few atoms (amplitude in **g** representing the longitudinal eigendisplacements) have properties unique to the global topology of the vibrational space, as opposed to the local chemical environment. Molecular dynamics simulations further explore the properties of the topological boundary modes under excitation and thermal fluctuations.

For electronic states, the bulk-boundary correspondence[91] is manifested in the exponential localisation of boundary modes. We now demonstrate that for the topological phases studied in this work, the eigenmode displacement $\varepsilon$ decays exponentially away from an arbitrarily placed heavier mass (Fig. 2a).

For an alternating strong/weak spring system with $N-1$ units of point masses $m_b$, and a single heavy unit of mass $m_H$, we can investigate the localisation of a boundary by choosing to express the eigenmode displacement of the unit of the chain furthest away from the boundary $v_{-1}$, as a function of $m_H$ and $N$. We find that the function can be expressed as:

$$v_{-1} = Ae^{-t(m_H)N} + B(N, m_H) \tag{1}$$

Consequently, the eigendisplacement furthest from the heavy boundary should vanish in the limit of large $N$,

$$\lim_{N \to \infty} v_{-1}(m_H, N) \to \delta; m_H > m_b \tag{2}$$

where $\delta$ is an asymptotic value related to $B(N)$. Supplementary Fig. 5 explores this exponential decay for the pSSH model. Equation (2) is the bulk-boundary correspondence expressed in a single unit. Figure 2 shows the bulk-heavy boundary correspondence for phonon SSH

analogue models studied further on. The principle relates the presence of a localised eigenmode next to the heavy unit when the winding number is different from 0. For the cases $\kappa_1 > \kappa_2$, with a strong spring next to the heavy unit, there is an exponential decay of the eigendisplacements along the chain when $m_H \geq m_b$. This topologically non-trivial phase is characterised by a winding number of 1 around the torus in the Brillouin zone (see Supplementary Fig. 3). Conversely, for the cases $\kappa_2 > \kappa_1$, with a weak spring next to the heavy unit, the eigenmodes are delocalised across the chain with an increment in the eigendisplacements away from the heavy mass. This trivial phase is characterised by a winding number of 0 (see Supplementary Fig. 3).

The above-mentioned point-mass pSSH model, consisting of a spring chain of alternating spring constants $\kappa_1$ and $\kappa_2$, is the prototypical model to study band inversion and the corresponding winding number under thermal fluctuations (Supplementary Fig. 3): Upon thermalisation and heavy boundary formation, a localised and mid-gap boundary mode evolves in the topological phase of the pSSH model (Supplementary Fig. 4). Similar results are also obtained for the fully-atomistic polyyne at a the DFTB level of theory (Fig. 2b and Supplementary Fig. 8), thus drawing a direct equivalence with the point-mass pSSH model. DFTB accurately predicts that the in-phase stretching of all triple bonds in $C_{50}H_2$ at 1840 cm$^{-1}$, falls within the expected range for Raman $\Gamma$-mode vibration for a chain of this length[92] in the range

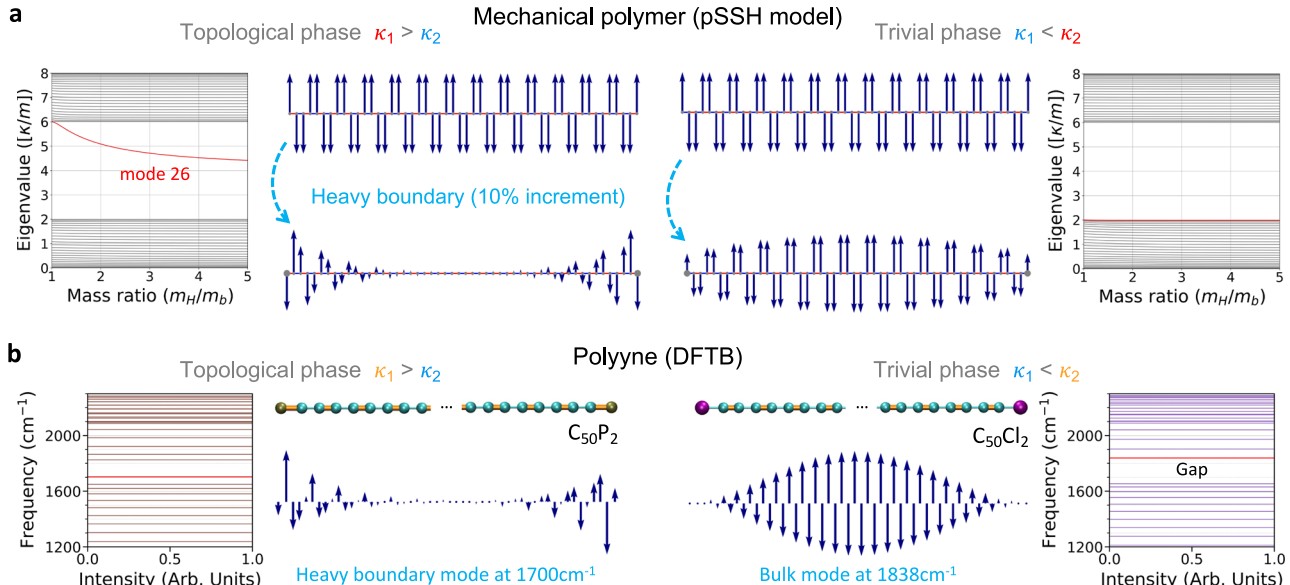

**Fig. 2 | Bulk-heavy boundary correspondence in phonon Su-Schrieffer-Heeger (SSH) analogues: point-mass polymer model vs. fully-atomistic polyyne density functional tight-binding simulation. a** An increase in the mass value at the boundary in a topological phonon SSH phonon analogue (pSSH) point-mass chain results in the expression of a vibration or phonon preferentially localised next to the heavy boundary. The example depicts the intensity of the longitudinal eigen-displacements (of eigenmode no. 26) for the topological (left hand side) and trivial (right hand side) cases before and after imposing a heavy-boundary condition for a pSSH system of 52 units. Mass ratio $m_H/m_b$ (heavy mass divided by bulk mass) increased from 1 to 1.1. Spring constant ratio $\kappa_1/\kappa_2 = 3$ (1/3) for the topological (trivial) case. Eigenvalue spectra as a function of the heavy mass are also shown (up to five-fold increase). Supplementary Figs. 5–7 elaborates on the bulk-heavy boundary correspondence for additional phonon SSH analogues. **b** Atomistic polyyne modelled using density-functional tight binding (DFTB) methods. The phosphaalkyne $C_{50}P_2$, featuring triple $P \equiv C$ bonds, realises a pSSH topological phase with a heavy-boundary mode at 1700 cm$^{-1}$ (left hand side), whereas the chloroalkyne $C_{50}Cl_2$, with Cl−C single bonds, is a trivial phase with no boundary mode. The longitudinal eigenmode displacements are depicted transversally for clarity.

10–200 K (Supplementary Fig. 8). To induce a pSSH-equivalent topological phase expressing a topological boundary mode, the hydrogen atoms at the termini of polyyne $C_{50}H_2$ are changed to heavier phosphorus atoms, which are triple-bonded to carbon ($C_{50}P_2$). In such a phase (cf. Supplementary Fig. 2), a mid-gap heavy-boundary mode exponentially localised next to the heavy phosphorus atoms is expressed. On the other hand, changing hydrogen in the trivial phase for a singly-bonded heavier atom, such as chlorine in the compound $C_{50}Cl_2$, does not provoke the expression of a boundary mode. While these results predict the existence of two phononic phases of matter in polyynes, they do not guarantee their experimental determination: polarons or large nuclear quantum effects may breakdown the harmonic approximations employed[93] or broaden phonon spectra to the point of making boundary modes undetectable.

### From 1D to 2D: the adsorbed SSH polymer chain

The 1D and quasi-1D pSSH studies are not representative of a polymer chain in solution or on a surface, due to the additional interactions and corresponding equivalent spring constants in these environments. To model a more realistic polymer environment, we introduce the adsorbed aSSH model on a surface, wherein an additional spring constant $\kappa_3$ fixes the chain to the surface (Fig. 3a, top panel). The aSSH polymer is modelled with equal masses, weak and strong alternating spring constants to account for the chemical bonds, and without parameterisation of van der Waals (vdW) interactions, angles or dihedrals (a full atomistic system is presented in the next section). Two decoupled types of longitudinal modes are recognised in the band structure, LA (LO) at lower (higher) energy. Furthermore, the absorption results in a transversal (T) flat mode (Fig. 3a, bottom panel). At the dynamical matrix level, the model aSSH has only time-reversal symmetry (see Supplementary methods). Moreover, the band inversion of the longitudinal modes renders a winding number of 1 around the torus in the Brillouin zone (see Supplementary Fig. 3). Upon heavy boundary formation, a

finite aSSH chain of 52 pearls expresses a TBM only when $\kappa_1 > \kappa_2$, namely the TBM is exponentially localised in the units next to heavy ones only when they are bonded by a strong spring (see Supplementary Fig. 6). The atomic surface roughness and height differences are ignored and thermalising the polymer at 100 K translates into serpentine motion of the polymer (Fig. 3b, top panel). Interestingly, the phonon band structure and specifically the acoustic band of the aSSH model changes during thermalisation (cf. Figure 3b,c). The last observation can be explained by the coupling with the out-of-plane (transversal T) resonators. Supplementary Fig. 9 shows how this coupling could occur: In the zigzag ($\theta = 15°$) aSSH model, there is the evolution of a new LA + T band due to the mixing with the T band mode. Figure 3c shows that the TBM in the aSSH model is expressed even in presence of disorder, without crystalline symmetry, owing to the intrinsic alternating spring constants. Here, MD simulations show that the TBM persists under thermal disorder. Note that the combination of heavy boundary formation and alternating spring constants express 'sublattice symmetry' or 'chiral symmetry'[94] (besides time-reversal symmetry) between the bottom (LA) and top (LO) bands, rendering the aSSH model under thermal disorder as BDI class[95,96]. Further, note that the LA + T mixing does not significantly change the band inversion, thereby paving the way for band inversion of few modes even in the presence of many resonators, as it would occur in a full-atomistic system. We perform normal mode analysis (NMA) of the linear aSSH model to understand the contribution of the transversal and longitudinal bands to the topological boundary state. We find that a heavy boundary expresses a longitudinal topological boundary mode (inset in Fig. 3d). It is fundamental to note that the topological boundary modes are expressed as (1) mid-gap states in the (2) second-last mass—two key indicators of the topological origin of the boundary modes and the bulk-boundary correspondence at play. The trivial case of the aSSH model does not show a TBM (Supplementary Fig. 10), owing to the presence of a weak spring next to the heavy boundary.

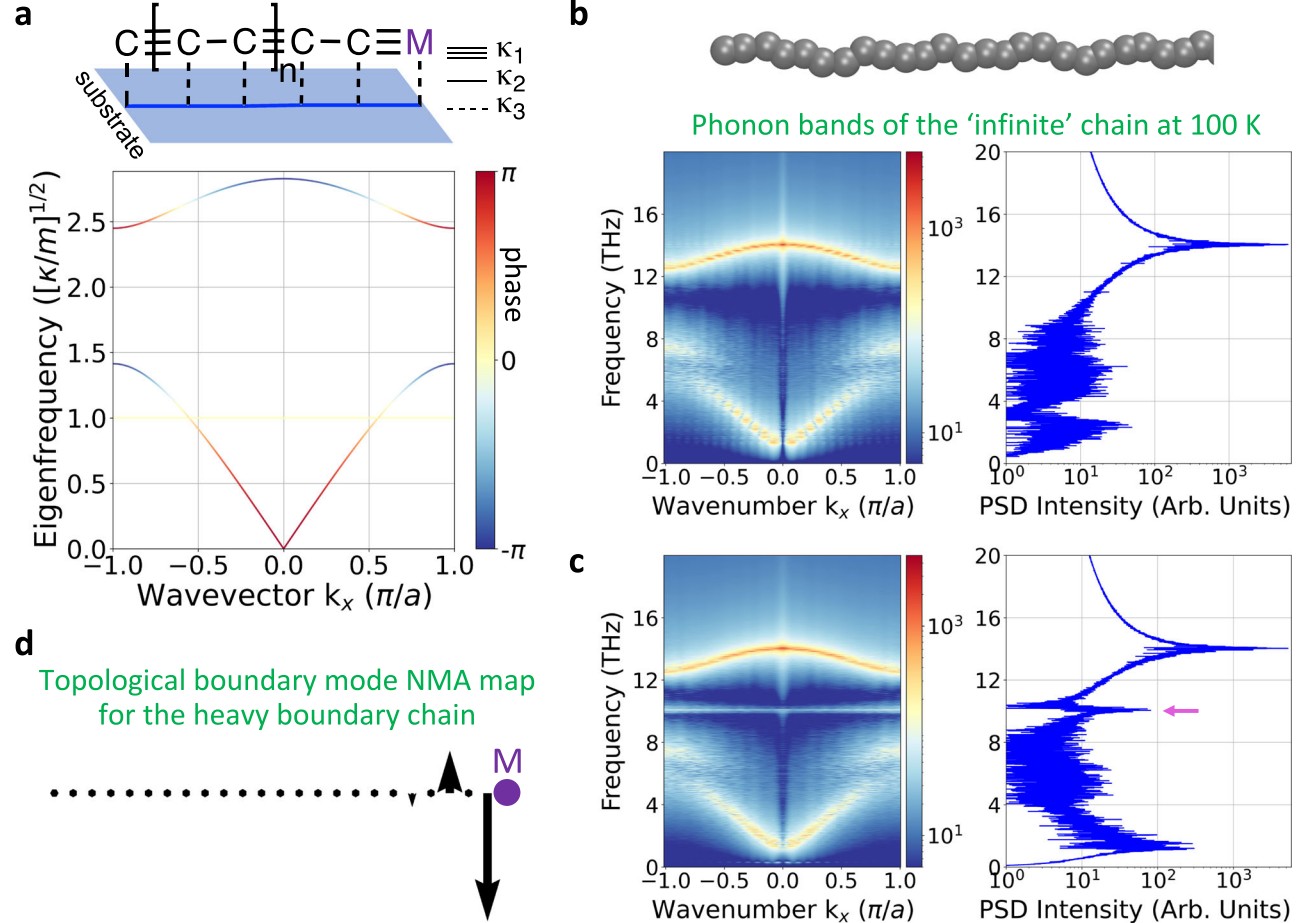

**Fig. 3 | The point-mass adsorbed Su-Schrieffer-Heeger (aSSH) model at equilibrium, polyyne–like equivalent under thermal fluctuations and their bulk-boundary correspondence. a** The aSSH model is realised by a four-parameter dynamical matrix: Mass $m_C$, $m_M$ and bonds $\kappa_1$, $\kappa_2$; and interaction with a virtual semi-unmovable substrate (blue line on the blue surface) via $\kappa_3$. This point-mass model overall mimics the molecular mechanics parameters of polyyne strongly adsorbed on a substrate (but without vdW parameters, angles nor dihedrals). Two masses per unit cell are allowed to move either parallel or perpendicular to the substrate. Here, the case of the transversal (T) mode crossing the lower energy mode (LA) is depicted in yellow. **b** Band dispersion from molecular dynamics simulations of an infinite-chain of 52 masses at finite temperature. Power spectral density (PSD) of the periodic chain shows a spectrum comparable to the point-mass mode. The influence of the T mode leads to a change in the LA band, see Supplementary Fig. 9. **c, d** A finite-chain with localised heavy-boundary masses expresses a longitudinal topological boundary mode (magenta arrow in **c**), whose eigenmode is depicted transversally in **d** for convenience. Source data are provided as a Source Data file.

## From 2D to 3D: the adsorbed SSH polymer double-chain

Having introduced the linear aSSH ($\theta = 0°$ Fig. 3a), zigzag aSSH ($\theta = 15°$, Supplementary Fig. 9) and random angle thermalised models (Fig. 3b, c), we now turn to a 'fixed zigzag' ($\theta = 60°$) aSSH model and its atomistic equivalent: A symmetry which can be realised by atomic or molecular chains self-assembled into arrays or supramolecular lattices. Figure 4 shows that the atomistic equivalent of the 'fixed zigzag' aSSH is a double chain connected by alternating effective strong and weak springs, which we refer to as the double-chain adsorbed SSH or daSSH model (Supplementary Fig. 11 shows that the difference in effective springs stiffness is due to adsorption asymmetry). The schematics of the daSSH are shown in Fig. 4a, whereby the dynamical matrix band structure is depicted in Fig. 4b. Two longitudinal acoustic (lower energy) and two longitudinal optical (higher energy) bands are recognized. One of each type presents phase inversion, also characterised by winding number of 1 around the torus in the Brillouin zone (see Supplementary Fig. 3). By using the parameters $\kappa_1 = 3$, $\kappa_2 = 1$, $\kappa_3 = 3$ and $\kappa_4 = \kappa_5 = 0.5$, $\theta = \alpha = 60°$, $m_M = 100$ and $m_C = 1$, the transversal mode (T) lies just above the lower LO band.

In the Supplementary Fig. 7, the exchange of $\kappa_1$ and $\kappa_2$ is further explored for a finite chain of 52 units, with heavy units, $m_M/m_C = 100$, at both edges. A double degenerate topological boundary mode is

exponentially localised in the units next to heavy ones only when they are bonded by a strong spring, meaning when $\kappa_1 > \kappa_2$. MD simulations at 100 K of the daSSH model of 52 masses reveals five main bands for the infinite-chain case (Fig. 4b) and TBM for the finite-chain case with localised heavy-boundary masses (Fig. 4c). The intensity of this TBM is mainly located in the mass next to the heavy one and decreases exponentially when moving to the bulk (Fig. 4d). As shown in Supplementary Fig. 11, the bands do not invert their phases for the trivial case of the daSSH model, leading to the absence of a TBM in the band dispersion from MD simulations at 100 K.

## Discussion

The expression of surface-confined topological phonon boundary modes under thermal fluctuations, paves the way for their engineering through molecular self-assembly, notably in the field of molecular materials on surfaces. The scope of their bulk-heavy boundary correspondence can be discussed in equivalent supramolecular systems, such as the daSSH model realised by two rows of an axial coordination lattice with substituted DABCO ligands on top of an iron porphyrin platform (Fig. 5a). Rows of the unsubstituted DABCO axial coordination lattice have been recently characterised by scanning tunnelling and atomic force microscopy[97] and choose a chiral trialkyl substituted

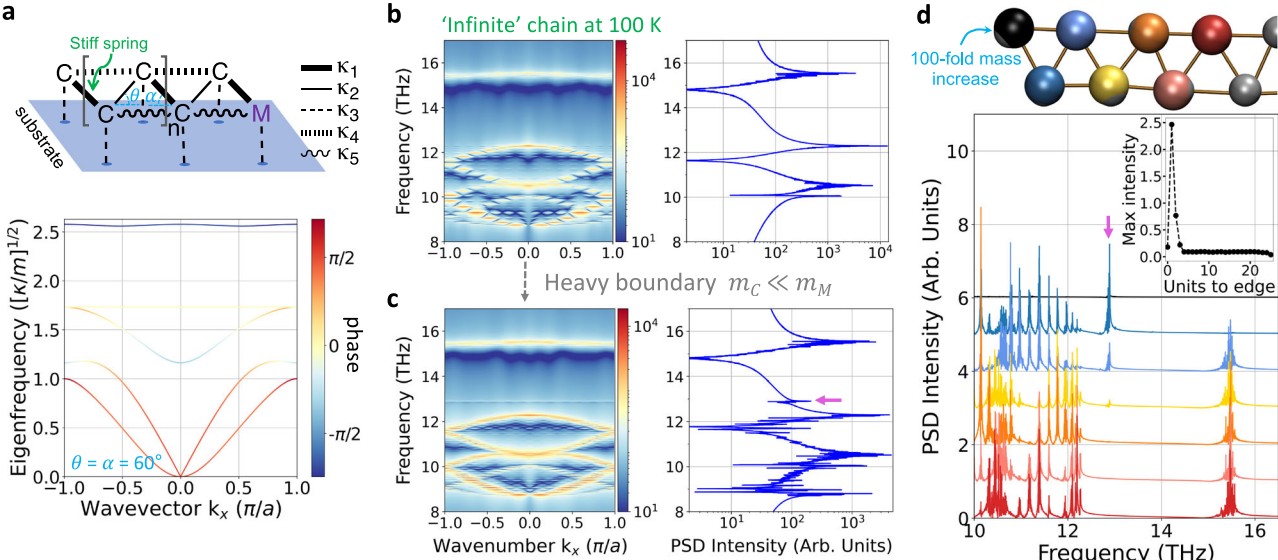

**Fig. 4 | A point-mass double-chain atomistic ladder Su-Schrieffer-Heeger (daSSH) model and atomistic equivalent under thermal fluctuations.** The model is realised by additional springs connecting next-nearer neighbour masses, leading to a dynamical matrix depending on the masses $m_C$, $m_M$ and angles $\theta$ and $\alpha$, and bonds $\kappa_1$, $\kappa_2$, $\kappa_4$, $\kappa_5$, with each mass bonded to a semi-unmovable substrate (blue dot on the blue surface) via a spring $\kappa_3$. **a** Following the dynamical matrix approach for the daSSH model in the case of $\theta = \alpha = 60°$, two decoupled types of modes are recognized: longitudinal, including two bands showing band inversion; and a transversal flat mode. **b** Band dispersion from molecular dynamics (MD) simulations of an infinite-chain of 52 masses at finite temperature considers more values of $\theta$ and $\alpha$ around the equilibrium value of 60°. **c** A finite-chain with localised heavy-boundary masses induces a topological boundary mode (TBM marked with magenta arrow) at 13 THz. **d** The boundary mode is mainly localised at the second-to-last mass (blue). The inset shows the maximum of the power spectral density (PSD) around the frequency of the TBM. Dashed lines serve as view guides when moving from the boundary to the bulk of the chain. A snapshot of the MD simulations shows the first seven masses of a finite chain. Source data are provided as a Source Data file.

DABCO (DABCO-3Bu) due to the simplicity of tight-packing alkyl chains by changing their length. The differences in spring constants are mimicked by the asymmetry of the underlying platforms leading to different vdW energies between and within rows, respectively: Supplementary Fig. 12 shows the inter- and intra-row dissociation curves. The resulting 'strong/weak' equivalent-spring pattern is shown in Fig. 5b. We also note that the chemical modification of the alkyl-chains or various axial ligands can enhance the asymmetry between 'strong/weak' SSH-like interactions. In this regard, alkyl chains help explore asymmetries at the weak interaction limit.

A mid-gap, second-last localised boundary mode (Fig. 5c) appears in the NMA simulation at the finite chain with heavy boundaries (Fig. 5f) which is absent in the infinite chain in Fig. 5d and the free boundary chain in Fig. 5e. A mid-gap signal is identified in the MD simulations at 10 K, further providing evidence of the plausibility of the daSSH-equivalent topological phase in supramolecular assemblies (Fig. 5f). The corresponding trivial case is constructed by an equivalent weak-spring termination (Supplementary Fig. 13), wherein no new boundary mode is identified upon creating the heavy boundary.

**The adsorbed SSH polymer double-chain in weakly-coupled supramolecular lattices**

We have developed analytical and heuristic guidelines for the expression of topological boundary modes in SSH chain models with the potential for experimental realisation employing supramolecular systems on surfaces. This procedure represents a first entry point for the design of boundary modes in lattice SSH models[98] formed by arrays of daSSH chains. Such supramolecular arrays can be designed by lattices consisting of alternating weak-strong hydrogen-bonded, metal-organic chains or van der Waals interactions. Here, three daSSH models derived from Fig. 5, containing 98 iron porphyrins and 98 DABCO-3Bu, are explored: A periodic boundary conditions (pbc) crystal (Fig. 6a) designed to illustrate vibrational properties not associated with boundaries; a ribbon (Fig. 6b) created by lifting the periodicity of the pbc crystal; and a heavy-boundary ribbon (Fig. 6c) after increasing the mass of the boundary molecules (coloured in red) of the ribbon three-fold. Figure 6d shows the NMA spectra for the pbc crystal and the heavy-boundary ribbon, depicting a SSH chain model boundary mode at 45.5 cm⁻¹ and localised between bulk modes. This boundary mode is exponentially localised at the second-to-last molecules as depicted in the NMA map of Fig. 6e. The phonon band structure from MD simulations at 10 K shows a small free-boundary mode appearing at 46 cm⁻¹ for the ribbon case (Fig. 6g) which is neither present in the pbc crystal (Fig. 6f), nor in the trivial case (Supplementary Fig. 14). A sharper signal is observed in the phonon spectra for the heavy-boundary ribbon case, (Fig. 6h) providing evidence of coupled daSSH topological boundary modes at finite temperature. Note that when coupling 1D topological phases, meticulous attention is necessary, as the classification might change significantly from quasi-1D chains to quasi-2D lattices[98]. Furthermore, patterning of boundary modes is possible by defining heavy masses anywhere in the material. Experimentally, such mass increase can be implemented through the use of heavier ligands or simply heavier isotopes. Figure 6i shows an X-shape periodic pattern of heavy masses located along the centre of a supramolecular ribbon, depicting the strong and weak interactions as defined in the equivalent-spring pattern in Fig. 5b. The NMA spectra for the nanoribbon (Fig. 6j) and the corresponding NMA map (Fig. 6k) show that the boundary mode at 20 cm⁻¹ is exponentially localized on the molecules adjacent to the heavy masses on the side of the strong interaction.

Further, the excitation profile of the plausible topological boundary mode is explored by applying an oscillating force to a single molecule during molecular dynamic simulations at 1 K[34]. The applied force is proportional to an element of the eigenmode depicted in Fig. 6e and using the corresponding excitation period of 0.733 ps (thermal equivalent of 65 K). The effect of this excitation is followed by means of the root mean square deviation (RMSD) from the starting minimised configuration. RMSD is used to identify fluctuations in the structural

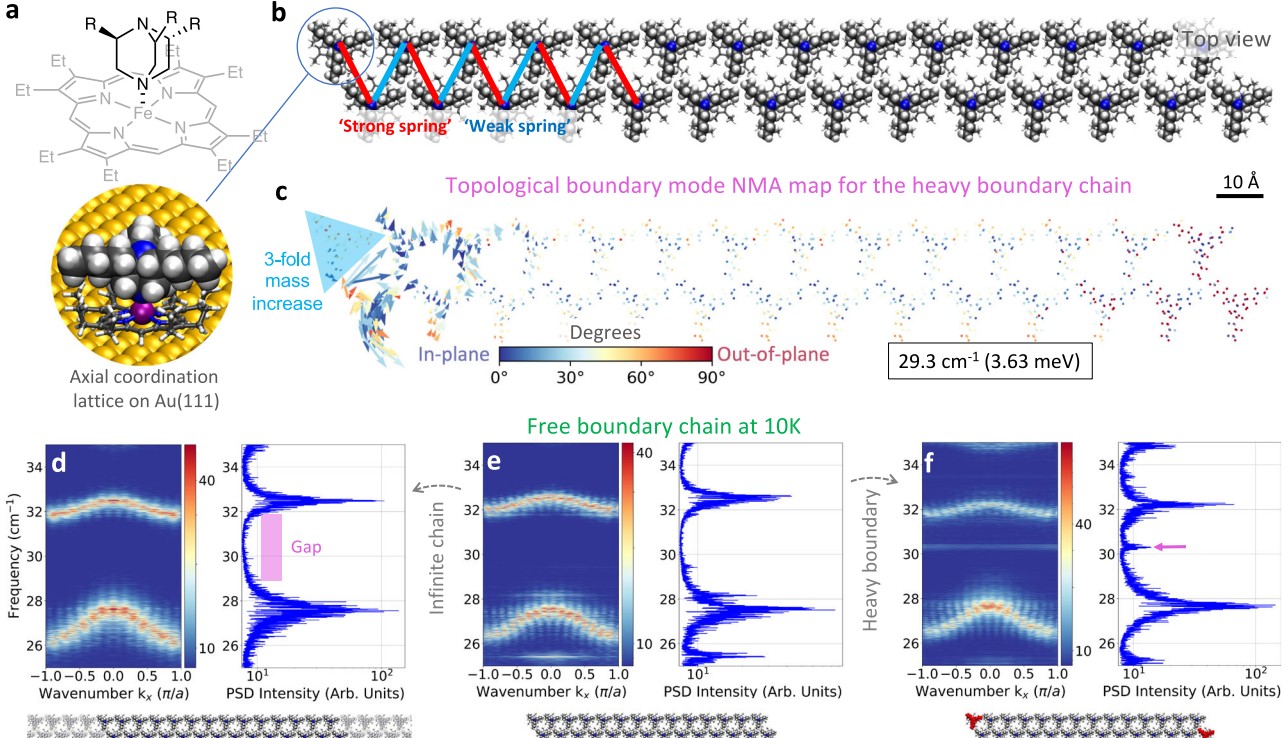

**Fig. 5 | An atomistic double-chain adsorbed Su-Schrieffer-Heeger (daSSH) model under thermal fluctuations. a** One nitrogen atom (blue sphere in 3D model) of a DABCO-3Bu molecule coordinates with the iron (purple sphere) of an octaethylporphyrin to form a supramolecular complex. **b** A double chain of the supramolecular complex is found equivalent to an alternating chain of strong and weak springs resembling the daSSH model. **c** Eigenmode mapping at 29.3 cm⁻¹ showing eigendisplacements exponentially localised at the second-to-last molecule from the heavy boundary of the finite chain. The colormap represents the angle of the eigenvector's components with respect to the $xy$-plane. **d–f** Band dispersions from molecular dynamics simulations at 10 K identify a (topological) boundary mode as a mid-gap signal. **d** The case with periodic boundary conditions (PBC) shows a gap between 28 and 32 cm⁻¹ in the power spectral density (PSD). Here, the shaded molecules depict the repetition of the main cell, represented by highlighted molecules. **e** Creating a boundary by lifting PBC (absent shaded molecules in d) and subsequently **f** a heavy boundary (heavier molecules marked in red), promotes the appearance of a signal assigned to a daSSH topological boundary mode at 30 cm⁻¹ (marked by a magenta arrow). Source data are provided as a Source Data file.

conformation of the molecules due to the excitation of a mode. The oscillating force is applied to the core of the red molecule circled in yellow in Fig. 7. Three replicas with different initial velocities were simulated. For each excitation case a baseline (without excitation) shares the same initial velocities. RMSD fluctuations show larger configurational changes in the excitation case compared to the baseline (Supplementary Fig. 15) for the neighbour and next-neighbour molecules. Moreover, the effect of the excitation shows a preference towards larger fluctuations in the molecules on the boundary, when compared to the bulk (cf. RMSD 2nd neighbour vs. 1st neighbour in Fig. 7).

Boundary modes that cannot be classified after the SSH chain model have been also identified in the supramolecular materials of Fig. 6⁹⁹. Figure 8 shows the phonon spectra in $y$ direction from MD simulations at 10 K. The lower frequency region of the band structure for the pbc crystal (Fig. 8a) shows a gap between 24 and 29 cm⁻¹, in which a very intense flat band appears at 26.1 cm⁻¹ (Fig. 8b). Analysis of the NMA intensity map for the ribbon (Fig. 8c) shows a prominent libration mode localised at the boundary of the material decreasing in intensity towards the bulk. The excitation profile of this libration boundary mode is also evaluated by applying an oscillating force to a single molecule (red molecule circled in yellow in Fig. 8d) during molecular dynamic simulations at 1 K. The force is applied to the one boundary molecule belonging to the eigenmode in Fig. 8c and using the corresponding excitation period of 1.249 ps (thermal equivalent of 38 K). Notably, the RMSD in Fig. 8d shows that the excitation does not propagate, contrary to the Fig. 7 and as compared to the RMSD baseline without excitation (Supplementary Fig. 16). Differences in

propagation between trivial and topological boundary modes are not an unambiguous marker for topology, but are a plausible consequence of topological phonons stemming from topological robustness against perturbation. Trivial and simply localised phonon boundary modes are disrupted by the phonon excitation, unlike topological boundary modes whose eigendisplacements are exponentially localised.

In summary, we studied phonon band formation on adsorbed polymer chain models and atomistic weakly-interacting molecular lattices, designed as SSH analogues with alternating soft and stiff springs, or spring-like interactions. Specifically, we showed that creating a heavy boundary in such polymer models and self-assembled supramolecular lattices (bulk-heavy boundary correspondence) leads to the emergence of exponentially localised phonons at the atom or molecule next to the heavy boundary for non-trivial topology. We show that such topological boundary modes can be patterned and are robust at finite temperatures, aiming towards phonon circuitry at the molecular scale. In supramolecular lattices, upon excitation of a single molecule of the mode, the modes stemming from the bulk–heavy boundary correspondence propagate, contrary to in-gap protected, free boundary modes. By demonstrating that topological boundary modes can be patterned by mass changes, we anticipate phonon engineering in self-assembled chains of strongly-interacting systems such as covalent, metal-organic, or supramolecular polymers and frameworks. Our work constitutes a stepping stone in topological phononics at interfaces, with far-reaching implications related to emerging quasiparticles unique to supramolecular systems,     and to phononic logic and eigensolvers in atomic-scale thermodynamic computing.

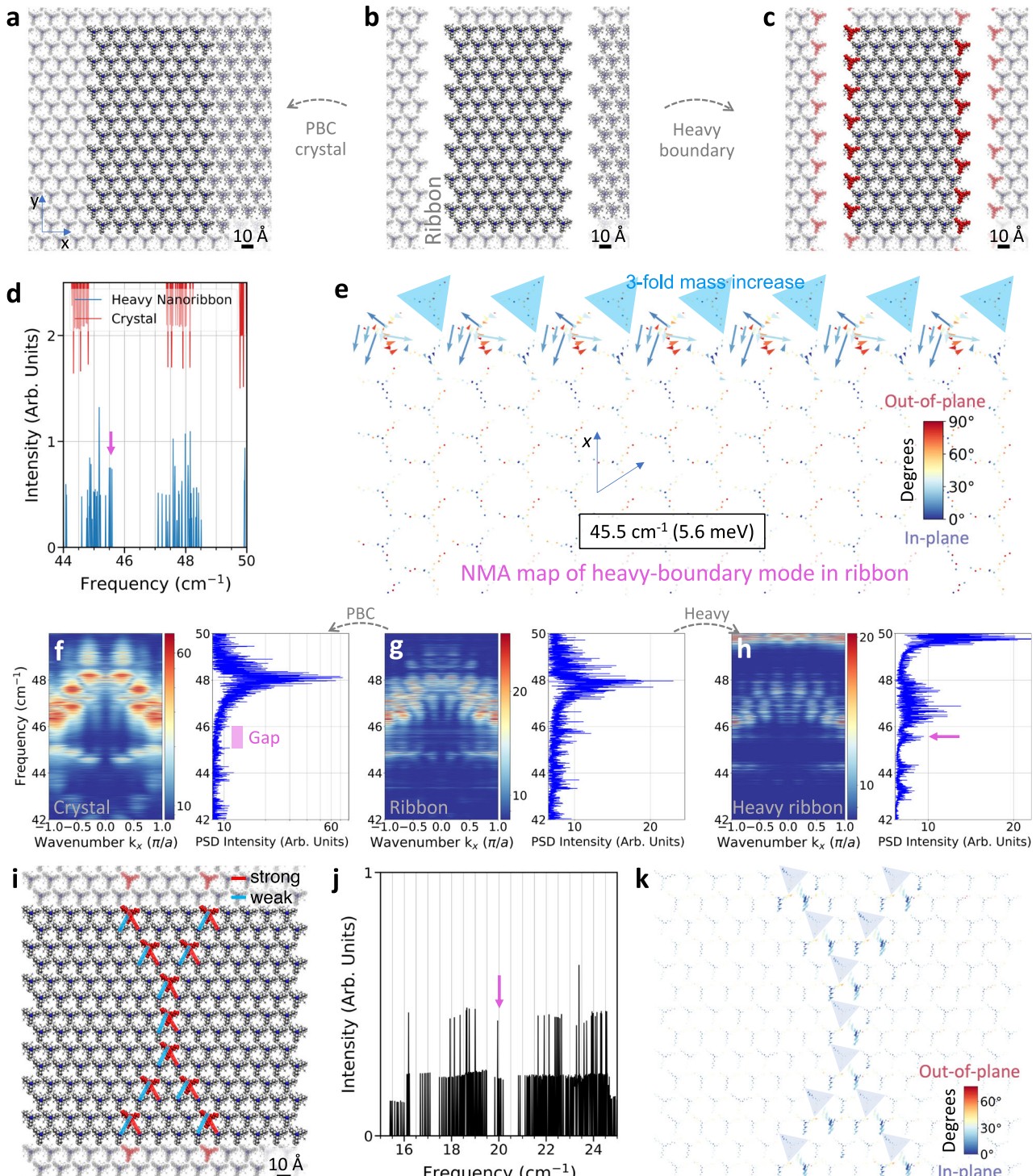

**Fig. 6 | A finite-ribbon double-chain adsorbed Su-Schrieffer-Heeger (daSSH)-array equivalent model with patterned heavy-boundaries under thermal fluctuations and guided excitation of the topological boundary mode.**
**a–c** Supramolecular array used to compare and pattern boundaries. Periodic boundary conditions (PBC) in $x$ and $y$ constitute a Crystal (**a**), PBC in $y$ form a Ribbon (**b, c**). Increasing the mass by three-fold in boundary molecules of the ribbon (in red) forms the heavy-boundary ribbon (**c**). **d** Normal mode Analysis (NMA) of Ribbon and Crystal shows a new eigenmode at 45.5 cm⁻¹ between bulk modes (magenta arrow), when the boundary of the crystal is opened. **e** Eigenmode mapping at 45.5 cm⁻¹ exhibits exponential localisation at the second-to-last molecules of the Ribbon. The colormap represents the angle of the eigenvector's components with respect to the $xy$-plane. **f–h** Band dispersions from molecular dynamics simulations at 10 K of the crystal (**f**), Ribbon (**g**) and Ribbon with heavy boundaries (**h**) helps to identify the sharp resonance of a potential topological boundary mode marked with magenta arrows. **i** Supramolecular Ribbon with heavier inner molecules, marked in red, whose mass has been increased three-fold. The shaded molecules depict the repetition of the main cell, represented by highlighted molecules. **j** NMA of the system showing an inner boundary eigenmode at 20 cm⁻¹ and whose mapping is shown in **k**. **k** Eigenmode mapping at 20 cm⁻¹, mostly localised at the molecules with a stronger interaction to the heavy ones as depicted in **i**. Source data are provided as a Source Data file.

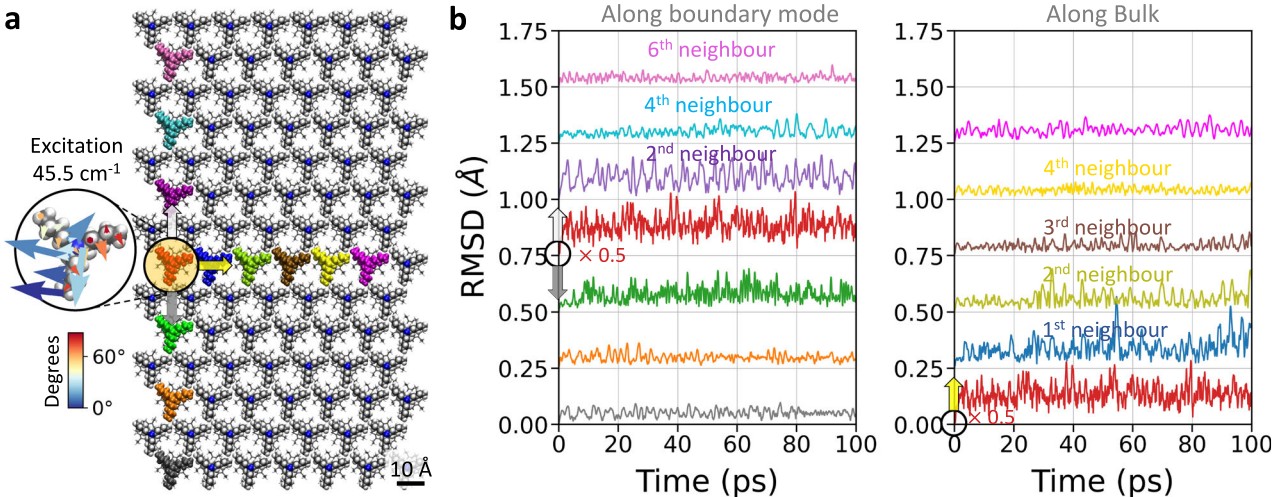

**Fig. 7 | Guided excitation of a boundary mode along the heavy-boundary.**
**a** During molecular dynamics simulations at 1 K, all non-hydrogen atoms of the molecule circled and highlighted in red are excited by an oscillating force (with a period of 0.733 ps or 45.5 cm$^{-1}$ and a force constant of 123.4 pN). The excitation occurs in the direction corresponding to the eigenmode of the single-molecule element depicted in Fig. 6e. The zoomed-in area shows the eigenvector's components of the excited molecule, whose angles with respect to the *xy*-plane are shown using the colormap. **b** The root mean square deviation (RMSD) from the initial minimised configuration for each molecule (with colours corresponding to those in panel **a**) shows fluctuations that propagate up to the 4th boundary neighbour. Each RMSD data set is offset by 0.25 Å relative to its (second) nearest neighbour along the (boundary) bulk for clarity. Same-seed trajectories without excitation are provided in the Supplementary Information Fig. 15. Arrows in panel **a** serve as a guide to the eye for the RMSD fluctuations in **b**. Source data are provided as a Source Data file.

## Methods

### Dynamical matrix for phonon band calculation

The connectivity matrices and corresponding dynamical matrix method for pSSH, aSSH and daSSH models is described in Supplementary Methods. Corresponding Python analysis scripts and Jupyter notebooks were employed to calculate the band structures (Figs. 2a, 3a and 4a and Supplementary Figs. 2, 3, 5, 6, 7, 9 and 11).

### Topological index

A winding number was calculated as the total phase difference gathered along a closed path in *k*-space[47]. The phase difference between the eigenstates *n* at two different points $k_0$ and $k_1$ is given by $\Delta\varphi_{n,1}$:

$$e^{i\Delta\varphi_{n,1}} = \frac{\langle u_n(k_0)|u_n(k_1)\rangle}{|\langle u_n(k_0)|u_n(k_1)\rangle|} \qquad (3)$$

$$\Delta\varphi_{n,\ell} = \text{Im}\ln\langle u_n(k_{\ell-1})|u_n(k_\ell)\rangle \qquad (4)$$

Where $u_n$ is the *n*th eigenvector with winding number given by:

$$w_n = \frac{1}{2\pi}\left(\text{Im}\sum_{\ell=1}^{N}\ln\langle u_n(k_{\ell-1})|u_n(k_\ell)\rangle + \text{Im}\ln\langle u_n(k_N)|u_n(k_0)\rangle\right) \qquad (5)$$

### Molecular simulations at finite temperature for phonon spectra calculation

Molecular dynamics for the calculation of the phonon spectra at finite temperature (Figs. 3b, c, 4b, c, 5d–f, 6f–h and 8a, b) and for NMA (in both models and atomistic systems) in this work were carried out using CHARMM c45b1[100] and Gromacs 2021.3[101]. Force field atom types and parameters were assigned from the CgenFF 3.0.1 force field[102,103]. Force fields reproduce the dynamics of large molecules with very high accuracy below 80 K[104] and can also reproduce probability densities[77]. Optimisations for the iron octaethylporphyrin (FeP) were used as reported by Adam et al.[105] and for DABCO by Burtch et al.[106]. A two-dimensional harmonic restraining potential was applied to the non-hydrogen atoms of the porphyrins core. Distance and orientation of

FeP are from Wang et al. from DFTB+ minimisation.[97] Simulations at finite temperatures were performed by means of the Bussi-Donadio-Parrinello thermostat with a 0.1 ps coupling constant. Configuration and force field files intended to reproduce the calculations and simulations are included in Supplementary Data. To obtain the phonon spectra (PSD) band we first calculate the projection of the velocities in the direction of a specific wavevector **k**:

$$\mathbf{v_k}(t) = \sum_i^N \mathbf{v}_i e^{-i\mathbf{k}\cdot\mathbf{r}_i(t)} \qquad (6)$$

Where $\mathbf{r}_i$ and $\mathbf{v}_i$ are the position and velocity of a specific atom in the *i*th unit cell, respectively, and *N* refers to the unit cells considered. Second, we calculate the autocorrelation function of $\mathbf{v_k}$ at a time *t* is given by:

$$C_k(t) = \frac{\langle\mathbf{v_k}(t_0)\cdot\mathbf{v_k^*}(t_0+t)\rangle}{\langle\mathbf{v_k}(t_0)\cdot\mathbf{v_k^*}(t_0)\rangle} \qquad (7)$$

Where $\mathbf{v_k}^*$ denotes the complex conjugate of $\mathbf{v_k}$. A more computationally efficient procedure to calculate the autocorrelation of $\mathbf{v_k}$ using fast convolutions with FFT algorithms can be used:

$$C_k(t) = F^{-1}\left[F_R(\mathbf{v_k})\cdot F_R^*(\mathbf{v_k})\right] \qquad (8)$$

where $F^{-1}$ is the inverse Fourier transform of the product of the Fourier transform of the velocity $F_R(\mathbf{v_k})$ with its complex conjugate $F_R^*(\mathbf{v_k})$. Finally, the PSD Is obtained as the discrete-time Fourier transform of $C_k$. As pointed out by Koukaras et al. this method should be employed for each atom type in the unit cell[107,108]: one $C_k$ for each atom type. For simulations in supramolecular systems, we averaged $C_k$ among alkyl carbons (types CT2 and CT3) to increase the sampling of the autocorrelation function.

### Density functional tight binding

Simulations at a quantum level were performed using the DFTB method as implemented in the 24.1 release of the DFTB+ software[109]. To fully relax the molecules to a local minimum, self-consistent DFTB

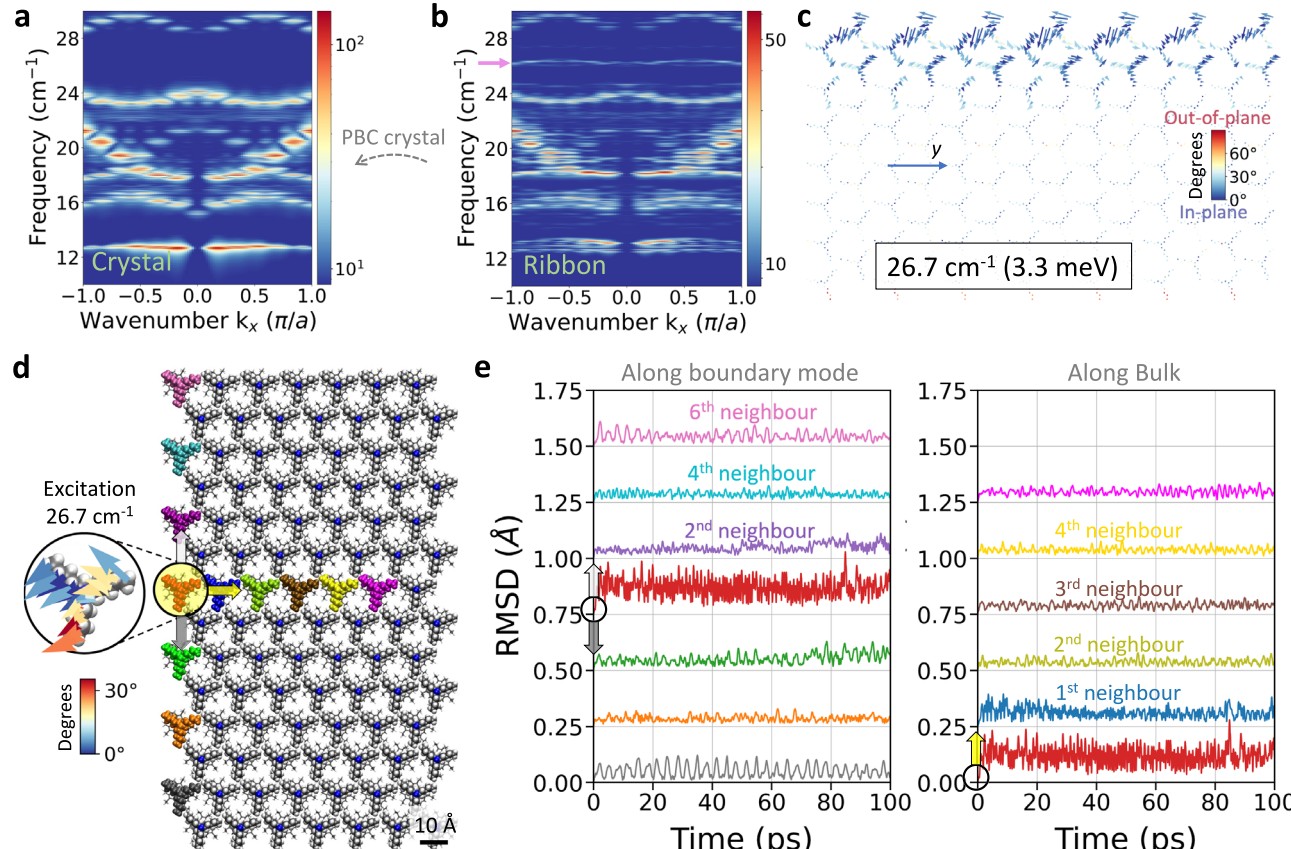

**Fig. 8 | Guided excitation of an in-gap free boundary mode. a**, **b** Band dispersion for molecular dynamics (MD) simulations at 10 K shows a gap between dispersive bulk nodes in the crystal material (**a**) and a localised boundary mode within this gap for the case of the Ribbon (**b**). **c** Eigenmode mapping at 26.7 cm⁻¹ localised at the edges of the Ribbon. The colormap represents the angle of the eigenvector's components with respect to the *xy*-plane. **d** All non-hydrogen atoms of the circled and coloured in red molecule are excited by an oscillating force (period of 1.249 ps or 45.5 cm⁻¹ and force constant 123.4 pN), in the direction corresponding to the eigenmode's single molecule element in **c** during MD simulations at 1 K. **e** The root mean square deviation (RMSD) from the initial minimised configuration for each molecule (with colours corresponding to those in panel **d**) reveals that the excitation does not propagate to the boundary neighbours. Each RMSD data set is offset by 0.25 Å relative to its (second) nearest neighbour along the (boundary) bulk for clarity. Same-seed trajectories without excitation are provided in the Supplementary Information Fig. 16. Arrows in panel **d** serve as a guide to the eye for the RMSD fluctuations in **e**. Source data are provided as a Source Data file.

calculations were performed with a tolerance set at $1 \times 10^{-6}$ electrons to ensure accurate electronic structure convergence. The parameters used to construct the Hamiltonian were derived from the Third-Order Parametrization for Organic and Biological Systems[110,111], which is known for its accuracy in modelling such molecular systems.

Normal modes were calculated using the mass-weighted Hessian matrix, calculated from finite difference second derivatives of the energy with respect to atomic positions. A finite difference step size of $10^{-6}$ atomic units was used.

### Molecular simulations at finite temperature for propagation studies

To explore the excitation profile at the nanoribbon's boundary, all the non-hydrogen atoms of a selected molecule were excited using an oscillating force during molecular dynamic simulations at 1 K, with the Berendsen thermostat in the parallel version of CHARMM c45b1[100] using 32 CPUs. The oscillating force is given by:

$$\mathbf{F}(t) = F_0 \cos\left(\frac{2\pi t}{T_k}\right)\mathbf{w}_k \qquad (9)$$

Where $\mathbf{w}_k$ is the partial eigenvector containing the entries of the eigenmode $k$ associated with the excited molecule, the rest of the entries are zero. $T_k$ is the period corresponding to the eigenvalue $\lambda_k$,

satisfying $\sqrt{\lambda_k} = 2\pi/T_k$. For the excitation in Fig. 5, $T_k = 0.7331$ ps (45.5 cm⁻¹) and $F_0 = 123.4$ pN, so that the maximum force applied to a single atom is limited to 50 pN. For the excitation in Fig. 6, $T_k = 1.249$ ps (26.7 cm⁻¹) and $F_0 = 123.4$ pN. To avoid thermalisation of the applied excitation, we set the scaling velocity frequency ieqfrq and auto-centering frequency ntrfrq (and iprfrq) to every 5 ps.

### Workflow for the identification of the topological boundary modes at finite temperatures

A standing challenge is the calculation of topological invariants from phonon band eigenvectors for materials with thousands of atoms and at finite temperature. Therefore, a workflow (Fig. 9) is employed to provide evidence of topological boundary modes for the materials in Figs. 5, 6 and 7 following the design principles of SSH heavy-boundary model introduced and elaborated in Figs. 3 and 4. We propose that this workflow can be used to find topological boundary modes in most materials which are designed following the aSSH or daSSH heavy-boundary models.

(a)  Normal mode analysis (NMA) using CHARMM c45b1[100] to obtain the spectra of independent harmonic oscillations.

(b)  Identification of boundary eigenvectors from NMA (Eigenmode maps in Figs. 5c, 6e and 8c) using custom Python scripts.

(c)  New modes detected after steps **a** and **b**, are verified to meet the following conditions: (i) are the new modes within a gap between

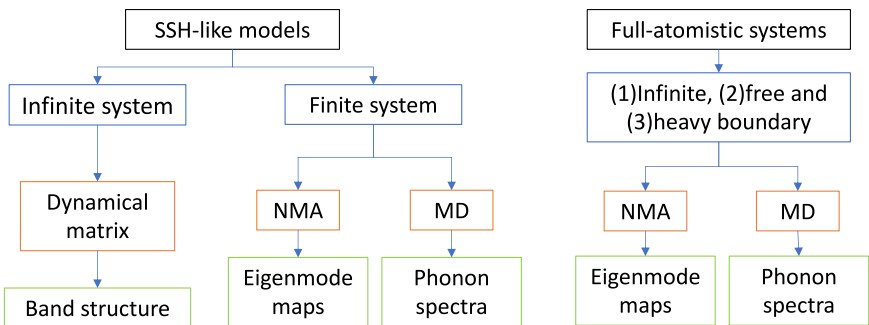

**Fig. 9 | Workflow for the identification of the topological boundary modes at finite temperatures.** Summary of the vibrational modes and phonon bands calculations performed. It is worth noting that the dynamical matrix approach is a one or two-dimensional model (see Supplementary Methods) and the normal mode analysis (NMA) and molecular dynamics (MD) methods involve three-dimensional force field calculations with the programmes CHARMM c45b1[103] and Gromacs 2021.3[104]. All MD simulations are performed at finite temperature.

bulk modes? (ii) are the new modes exclusively localised at second-to-last molecules? Now these modes are candidates to TBM.

(d) If the candidates are also found in the PSD band from MD simulations at finite temperatures, then they are called boundary modes.

(e) Finally, the topology origin of the modes is evidenced by constructing counter-examples: (1) trivial case and (2) the non-heavy, free boundary example and repeating steps **a**–**d**. Both counter examples should not fulfil the criteria of the candidate TBM at similar energies, that is, should not express an in-gap boundary mode, should not be localised at second-to-last molecule, nor should be found in the band dispersion at finite temperatures.

## Source Data
Source Data file including all the numerical source data for the plots.

## Data availability
The authors declare that the data supporting the findings of this study are available within the paper and its Supplementary Information files. Source data are available with this paper. Source data are provided with this paper.

## Code availability
Simulation files, Python analysis scripts and Jupyter notebooks have been deposited in the NOMAD database under accession code https://doi.org/10.18653/v1/2020.acl-main.173https://doi.org/10.17172/NOMAD/2024.11.20-1.

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

## Acknowledgements

This work was partly funded by the National Science Foundation of China, the Chinese Academy of Sciences (nos. QYZDBSSW-SLH038, XDB33000000 and XDB33030300) and the Deutsche Forschungsgemeinschaft (DFG, German Research Foundation) Cluster of Excellence 'Matters of Activity. Image Space Material' (no. 390648296). We gratefully acknowledge financial support from the Alexander von Humboldt Foundation (C.-A.P.). We thank Max Ünzelmann, Hibiki Orio, Friedrich Reinert and Hongming Weng for fruitful discussions. We acknowledge support by the Open Access Publication Fund of Humboldt-Universität zu Berlin.

## Author contributions

J.D.C.G. performed the simulations, developed the methodology and analysed the data. J.R., K.P. and J.P.R. discussed and co-supervised the project. C.-A.P. analysed the data, developed the methodology, designed the research and supervised the project. All authors commented on the manuscript.

## Funding

## Competing interests

The authors declare no competing interests.
