## [Transparent Peer Review file · Nature Communications]

Dynamics and patterning of topological phonon boundary modes in polymer chains and supramolecular lattices on surfaces

Corresponding Author: Professor Carlos-Andres Palma

This manuscript has been previously reviewed at another journal. This document only contains reviewer comments, rebuttal and decision letters for versions considered at Nature Communications.

Version 1:

Reviewer comments:

Reviewer #1

(Remarks to the Author)

In the paper, Gonzalez et. al. study the possibility to realized phonetic SSH-like models in polymer chains and supramolecular lattices.

The 1D SSH model is one of the prototypical models demonstrating non-interacting topological order. Due to a chiral symmetry that anti-commutes with the hamiltonian, one can define an bulk topological invariant that is quantized to integer values. The value of the invariant is directly related to the appearance of topological corner states, and therefor one of the most simple models showing a topological phenomenology.

Traditionally, the SSH model is formulated in terms of fermions (electrons). Over the last few years, it has become apparent that one can not only topologically characterize electronic hamiltonians, but also the dynamical matrix appearing in phononic (bosonic) problems.

This phenomenology and other topological models have previously been realized in mechanical systems, however the study of topological phonons in real materials has remained less explored. There are papers that classify and enumerate possible phononic topological phases (e.g. arXiv:2211.11776), however experimentally there has been little work compared to acoustic, electronic and photonic topological materials.

Therefore the idea to find material realizations of phononic topological materials is an interesting one to me. The authors propose that one can find and realize topological SSH like states in polymer chains and 2D supramolecular lattices, which are well studied and characterized experimentally. The paper explores shows that a simple model can account for the topological boundary modes in these systems and molecular dynamics calculations show that the topological states are stable with respect to small-to-moderate temperatures.

The authors further design a 2D-like supramolecular lattice consisting of SSH-like coupled chains, that also shows the phenomenology expected from a topological surface states.

I find that the paper is very interesting and worth publishing. It is also of interest to the larger community of Nat. Comm., due to the explicit treatment of realistic molecular systems and experimentally measurable consequences.

I have some comments that the authors should think of addressing.

1) I find the introduction hard to read and understand at times. To me, the authors use a lot of technical jargon without much content. Technically speaking they are not wrong, however I think they are doing themselves a disservice, because readers new to this field will have trouble understanding.

Examples are:

“topology in physics studies the non-Euclidean spaces and their topology of equations of motions, usually in the framework ff

matrix operators and band theory”

“distinct elements in the holonomy group of the vibrational space [17]”

I urge the authors to rephrase this in a non-technical and property focused way. There are a few more examples.

They also cite ref.17, which is a paper by some of the authors that has been recently submitted, however it is not accessible to the reader. These things confuse more than they do good.

2) The authors refer to figures in the SI and extended data. As a reader, I do not want to have 3 pdfs open to understand what the text is about. If it is necessary to be mentioned explicitly in the main text, the figure belongs into the main text. It makes it very tedious and hard to read.

3) The authors discuss the SSH model a lot, but they do not go into the (very well explored) topological classifications. As I mentioned above, the classical SSH model is protected by Chiral symmetry (Class AIII), and the invariant is a winding number that has to be calculated along a one-dimensional path. It is well known that one can break chiral symmetry and still get topological modes, however this requires inversion symmetry [see Nature 547, 298–305 (2017)]. However now, the classification is that of obstructed atomic limits, and weaker than the classical winding number classification.

I have a hard time believing that the polymer chain or the double chain model have a chiral symmetry in a realistic setting. A substrate will also break inversion symmetry, so what is protecting these modes? Are the examples just fine-tuned, i.e. in a case of a different surface or material, the boundary states will merge into the bulk states?

Further, the authors discuss a 2D-like SSH model (Fig. 5). When coupling 1D topological phases, one has to be very careful, as the classification might change significantly. Take e.g. the 2D SSH model, which has very different topological phases than the 1D model. For example, it cannot really be defined using a winding number. Further, the most famous variant of the 2D SSH model, the BHZ model, finds a so-called fragile topology, which is characterized by corner states instead of 1D boundary modes. See e.g. Science 357,61–66 (2017) and PRB 107, 045118 (2023). What that means is that the authors need to be more precise to help the reader understand which type of topology they are talking about. Right now, this is unclear to me. This will help to substantiate the findings of the paper and will increase its possible impact.

Reviewer #2

(Remarks to the Author)

The authors studied the topological phonon boundary modes in polymer chains and supramolecular lattices on surfaces.

Due to the following reasons, I can not accept this work.

1. Major thing: The authors stated that 'The study of topology in the context of electronic phases of matter is widespread. Yet vibrational and dynamical topological phases of the matter remain to be explored'. I do not agree this comment. Please note that the topological phonon phases have been widely investigated in solids and artificial crystals, see Physical Review Letters, 2016, 117(6): 068001, Advanced Functional Materials, 2020, 30(8): 1904784., Proceedings of the National Academy of Sciences, 2016, 113(33): E4767-E4775., Science Advances, 2020, 6(46): eabd1618., Physical Review Letters, 2023, 131(11): 116602, etc... I can list more than 50 papers focusing on topological phonons. The vibrational and dynamical topological phases of matter are not new, and the study is also widespread. Therefore, this is the main reason I must reject this work, I think communications physics is a good home for the current paper.

Note that compared with the model for polymer chains and supramolecular lattices, the topological phonons study of 3D realistic materials may be more useful for the follow-up experimental studies. Hence, again, I do not suggest this work be published in NC now; it may be ok three years ago, not now.

2. Minor thing: The authors mentioned many times about the topological boundary mode. However, the authors did not prove the boundary mode is topologically nontrivial. At least you can understand the boundary based on the nontrivial topological index to show the readers that the boundary is topologically nontrivial.

Reviewer #3

(Remarks to the Author)

The manuscript describes a series of Su-Schrieffer-Heeger (SSH) phonon models and their topological features. Specifically, the models discussed are the traditional chain, a chain adsorbed on a substrate, a double chain adsorbed on a substrate, and a supramolecular lattice. These models are then associated with real polymers and derived materials, and their finite temperature properties simulated using molecular dynamics. The various topological edge modes are characterized, and a specific prediction is their propagation upon the excitation of single molecules.

While topological phonons is an interesting field of research, it is unclear that Nature Communications is the best journal for this work. There is a significant body of work on topological phonons (which the authors seem to be unaware of), and it is unclear that the results presented here provide any fundamentally new insights or discover novel phenomena associated with topological phonons. In particular:

1. The authors write: "Thus far, mostly mechanical models, such as Maxwell lattices, have been employed to study vibrational topological properties". In fact, there is a very large body of literature exploring topological phonons in atomistic systems both theoretically and experimentally, with a non-exhaustive list including:

* One of the early experimental observations of topological phonons: <https://scholar.google.com/citations?>

view_op=view_citation&hl=zh-CN&user=thgksIEAAAAJ&citation_for_view=thgksIEAAAAJ:zYLM7Y9cAGgC

* Experimental observation of topological phonons in graphene:

<https://journals.aps.org/prl/abstract/10.1103/PhysRevLett.131.116602>

* And a recent full catalogue of topological phonons in thousands of materials: <https://arxiv.org/abs/2211.11776>

* Database associated with the above work: <https://www.topologicalquantumchemistry.fr/topophonons/>

In this context, the present work provides yet another characterization of topological phonons in atomistic systems. It is unclear whether there is something special about the systems studied in this work compared to many earlier works (and I emphasize again that the list above is by no means exhaustive).

2. On the methodological front, can the authors please expand on the meaning of MD simulations at 1K or 10K? Wouldn't the dynamics of the system at that temperature be dominated by quantum nuclear effects, which as far as I understand are not included in the calculations presented?

3. Can the authors expand on the propagation vs. non-propagation nature of boundary modes? Is this a completely unambiguous marker of topological phonons?

4. More generally, could the authors clarify what is special about topological phonons in these particular materials that merits the report of this work in Nature Communications? Are there new fundamental insights into topological phonons? Or are there particular applications that this class of material would enable but others wouldn't? Something else?

Version 2:

Reviewer comments:

Reviewer #1

(Remarks to the Author)

I think the authors have addressed the previous shortcomings in a satisfactory way. Because this is a new material platform and because it might bridge topological materials and the more chemically oriented communities, I think the manuscript should be published.

Reviewer #3

(Remarks to the Author)

I thank the authors for considering my questions and suggestions. Unfortunately, I still think that Nature Communications is not the best venue for this work, and likely a more specialized journal addressing the soft matter community is more adequate. To support my recommendation, the main point is:

1. It is still unclear to me that the authors reveal any new aspect of topology that arises because of the use of polymers. Instead, it appears that they simply describe well-understood topological features in yet another family of materials. I do not think that this merits publication in Nature Communications, but instead in a more specialized journal for the polymers community.

But also:

2. I am not convinced by the response of the authors regarding the use of MD simulations at 1K or 10K. I am not surprised that rotational degrees of freedom are well described below 80K, as these are very low energy modes whose quantum fluctuations are tiny. My main worry was the higher energy modes (e.g. C-C bond vibrations), whose quantum fluctuations are large, and can dominate up to very high temperatures, certainly at 1K or 10K. The authors should clarify this point further.

Version 3:

Reviewer comments:

Reviewer #3

(Remarks to the Author)

I thank the authors for considering my comments again. I still think that the work will be of interest mostly to the community working on polymers and related compounds, rather than the wider community working on topology and beyond. However, I understand that others may have different views on this.

Reviewer #3 (Remarks to the Author):

I thank the authors for considering my questions and suggestions. Unfortunately, I still think that Nature Communications is not the best venue for this work, and likely a more specialized journal addressing the soft matter community is more adequate. To support my recommendation, the main point is:

1. It is still unclear to me that the authors reveal any new aspect of topology that arises because of the use of polymers. Instead, it appears that they simply describe well-understood topological features in yet another family of materials. I do not think that this merits publication in Nature Communications, but instead in a more specialized journal for the polymers community.

We very much appreciate the reviewer's time and comments.

Our study is foundational in its application to synthetic, atomistic soft-matter, which is not only "yet another family of materials". To clarify the importance of polymers and synthetic organic chemistry to this end, we have now extended the introduction as follows:

"Yet vibrational and related dynamic and phonon topological phases, especially at finite temperatures and in the presence of disorder, remain to be explored in atomistic soft matter i.e. molecule-based materials and synthetic materials such as polymers, self-assembled networks, metal-organic frameworks, and covalent organic frameworks, to mention a few. Phonon bands are a universal property of many extended molecule-based materials, a highly diverse set of materials which can be built from millions of compounds. Thus, the prospect of chemical compounds belonging to two different sets of phonon phases of matter—a topological phase and non-topological phase—could bear far-reaching implications."

We do not fully consider that topological phonons are "well-understood" in real atomistic materials, not even regarding the prototypical SSH model: To the best of our knowledge, we study the effect of mass in the localization of the boundary modes for the first time, and in realistic materials. To substantiate the accuracy of this statement we searched Scopus for the keywords "phonon" & "heavy" & "topology" and found 14 matches, none of which pertinent to the bulk-heavy boundary correspondence. More importantly we extend the 1D SSH model to 2D and 3D surface-confined models and under thermal fluctuations.

But also:

2. I am not convinced by the response of the authors regarding the use of MD simulations at 1K or 10K. I am not surprised that rotational degrees of freedom are well described below 80K, as these are very low energy modes whose quantum fluctuations are tiny. My main worry was the higher energy modes (e.g. C-C bond vibrations), whose quantum fluctuations

are large, and can dominate up to very high temperatures, certainly at 1K or 10K. The authors should clarify this point further.

We have now made it clear how electronic effects and quantum corrections are important in the introduction:

'In order to predict topological phonon phases in soft-matter, at least four steps are needed: Revision or development of mathematical topological models which capture key symmetries and dimensions, addressing the effect of disorder and temperature on said models, followed by assessment of electronic and quantum corrections.'

Therefore, before moving to electron-phonon or quasiparticle dressing, and corresponding vibrational quantum correction benchmarking, we performed studies considering electronic effects at the density functional tight binding theory molecular dynamics level up to 200 K (DFTB MD) on chemically plausible polymers. (Note that the in-phase stretching of all triple bonds of $C_{50}H_2$ at 1842cm^{-1} predicted by DFTB falls within the expected range for Raman Γ -mode vibrations for a chain of this length; e.g. *Physical Review B* 94, 195422, 2016).

We have included these relevant results in Figure 2 and Supplementary Figure 3, below.

Figure 2. Bulk-heavy boundary correspondence in phonon SSH analogues: Point-mass polymer model vs. fully-atomistic polyynes density functional tight-binding simulation

a. An increase in the mass value at the boundary in a topological phonon SSH point-mass chain results in the expression of a vibration or phonon preferentially localized next to the heavy boundary. The example depicts the intensity of the longitudinal eigendisplacements (of eigenmode no. 27) for the topological (left hand side) and trivial (right hand side) cases before and after imposing a heavy boundary condition for a pSSH system of 52 units. Mass ratio m_H/m_b increased from 1 to 1.1. $\kappa_1/\kappa_2 = 3$ (1/3) for the topological (trivial) case. Eigenvalue spectra as a function of the heavy mass are also shown (up to five-fold increase). **Extended Data Figure 2-4** elaborates on the bulk-heavy boundary for additional phonon SSH analogues. **b.** Atomistic polyynes modelled using DFTB. The phosphoalkyne C_{50}P_2 , featuring triple $\text{P}\equiv\text{C}$ bonds, realises a pSSH topological phase with a heavy boundary mode at 1700 cm^{-1} (left hand side), whereas the chloroalkyne C_{50}Cl_2 , with $\text{Cl}-\text{C}$ single bonds, is a trivial phase with no boundary mode. The longitudinal eigenmode displacements for these pseudo-1D systems are depicted transversally for clarity.

Supplementary Figure 3. Normal mode analysis (NMA, **a-c**) and power spectra (PSD, **d-f**) for 1D polyynes chains, consisting on 50 carbon atoms with H (**a,d,g**), Cl (**b,e,h**) and P (**c,f,i**) as termini atoms. Simulations at a quantum level were performed using DFTB. The in-phase stretching of all triple bonds of $C_{50}H_2$ at 1840cm^{-1} (**a**) is reproduced in the PSD at 10 K (**d**), within the expected range for Raman Γ -mode vibration for a chain of this length⁶. Both NMA and PSD for $C_{50}H_2$ and $C_{50}Cl_2$ show a gap between acoustic and optical branches. When a heavier atom such as phosphorus is attached via triple bond to the carbon chain, $C_{50}P_2$, a heavy boundary mode appears within the gap. The inset in **c** shows that the eigendisplacement (depicted transversally for convenience) of the boundary modes (numbers 131 and 132) is exponentially localized in the mass next to the heavy one. The boundary mode in $C_{50}P_2$ can be identified at temperatures between 10 and 200 K (**g**). Snapshots after 100 ps of DFTB MD simulations at 200 K are depicted as insets in **g** ($C_{50}H_2$), **h** ($C_{50}Cl_2$) and **i** ($C_{50}P_2$).

Our studies with DFTB up to 200 K are a first step towards investigations with Ehrenfest DFTB methods (*Journal of Chemical Theory and Computation* 16, 4454-4469, 2020), and towards performing corresponding benchmarks to Ehrenfest DFT methods (e.g. *Physical Review X* 7, 031035, 2017) and to current approaches for quantum fluctuation corrections (e.g. *Journal of Chemical Physics* 152, 124104, 2020). We have submitted two DFG grants to perform these benchmarks. Until then, we acknowledge that it is currently unknown if an explicit treatment of the quantum dynamics of the nuclei on the electronic ground state potential energy surface would enhance or suppress the expression of topological boundary modes. Therefore, we write in the main text:

“While these results predict the existence of two phononic phases of matter in polyynes, they do not guarantee their experimental determination: polarons or large nuclear quantum effects may breakdown the harmonic approximations employed or broaden phonon spectra to the point of making boundary modes undetectable” .

Reviewers' comments:

Reviewer #1 (Remarks to the Author):

In the paper, the authors study the possibility to realized phonetic SSH-like models in polymer chains and supramolecular lattices.

The 1D SSH model is one of the prototypical models demonstrating non-interacting topological order. Due to a chiral symmetry that anti-commutes with the hamiltonian, one can define an bulk topological invariant that is quantized to integer values. The value of the invariant is directly related to the appearance of topological boundary states, and therefore one of the most simple models showing a topological phenomenology.

Traditionally, the SSH model is formulated in terms of fermions (electrons). Over the last few years, it has become apparent that one can not only topologically characterize electronic hamiltonians, but also the dynamical matrix appearing in phononic (bosonic) problems.

This phenomenology and other topological models have previously been realized in mechanical systems, however the study of topological phonons in real materials has remained less explored. There are papers that classify and enumerate possible phononic topological phases (e.g. arXiv:2211.11776), however experimentally there has been little work compared to acoustic, electronic and photonic topological materials.

Therefore the idea to find material realizations of phononic topological materials is an interesting one to me. The authors propose that one can find and realize topological SSH like states in polymer chains and 2D supramolecular lattices, which are well studied and characterized experimentally. The paper explores shows that a simple model can account for the topological boundary modes in these systems and molecular dynamics calculations show that the topological states are stable with respect to small-to-moderate temperatures. The authors further design a 2D-like supramolecular lattice consisting of SSH-like coupled chains, that also shows the phenomenology expected from a topological surface states.

I find that the paper is very interesting and worth publishing. It is also of interest to the larger community of Nat. Comm., due to the explicit treatment of realistic molecular systems and experimentally measurable consequences.

I have some comments that the authors should think of addressing.

1) I find the introduction hard to read and understand at times. To me, the authors use a lot of technical jargon without much content. Technically speaking they are not wrong, however I think they are doing themselves a disservice, because readers new to this field will have trouble understanding.

Examples are:

“topology in physics studies the non-Euclidean spaces and their topology of equations of motions, usually in the framework ff matrix operators and band theory”

“distinct elements in the holonomy group of the vibrational space [17]”

I urge the authors to rephrase this in a non-technical and property focused way. There are a few more examples.

They also cite ref.17, which is a paper by some of the authors that has been recently submitted, however it is not accessible to the reader. These things confuse more than they do good.

We thank the reviewer for the comments. We have now provided a general introduction to topology from at least three approaches: The algebraic approach, the differential geometry approach, and the study of band crossings in the elementary band representation approach. The mentioned reference has been uploaded with this submission.

2) The authors refer to figures in the SI and extended data. As a reader, I do not want to have 3 pdfs open to understand what the text is about. If it is necessary to be mentioned

explicitly in the main text, the figure belongs into the main text. It makes it very tedious and hard to read.

We agree and we are committed to making our text self-contained.

For example, we have now included a new Figure in the main text which compares trivial vs. nontrivial phases for a broad audience. We also plan to move all instances of SI referrals into the methods section. We have not done this yet to facilitate tracking the latest modifications. We are committed to improving the self-containment of our work before publication.

3) The authors discuss the SSH model a lot, but they do not go into the (very well explored) topological classifications. As I mentioned above, the classical SSH model is protected by Chiral symmetry (Class AIII), and the invariant is a winding number that has to be calculated along a one-dimensional path. It is well known that one can break chiral symmetry and still get topological modes, however this requires inversion symmetry [see Nature 547, 298–305 (2017)]. However now, the classification is that of obstructed atomic limits, and weaker than the classical winding number classification. I have a hard time believing that the polymer chain or the double chain model have a chiral symmetry in a realistic setting. A substrate will also break inversion symmetry, so what is protecting these modes? Are the examples just fine-tuned, i.e. in a case of a different surface or material, the boundary states will merge into the bulk states?

Thank you. The examples are not finely-tuned in terms of different materials. To substantiate this, we now discuss the bulk-boundary correspondence in detail, describing the topological boundary modes in the pSSH, aSSH and daSSH models with a wide range of material parameters, see below, new **Figure 2** and **Extended Data Figure 2,3**.

The SSH-analogue model is known to have SPT (symmetry protected topology) classification e.g. BDI (ref. 46, PNAS 113, E4767–E4775, 2016). For the pSSH and daSSH models, the topological phase of the models is now classified by its band winding number. We believe this is sufficient, since alternate classification of the aSSH and daSSH e.g. as SPT classification, would indeed require extensive investigation of limiting cases for surface parameters, and new exploration of realistic modelling beyond their symmetries as the reviewer suggests. For example, realistic adsorption might require a description of commensurability which could enlarge parameter space (different spring constants) in the direction of the surface. Therefore, the reviewer is correct that in the adsorbed models, boundary states could disappear as the adsorption becomes more realistic, but such are limiting cases whose further study we motivate in our current manuscript.

Supplementary Figure 1. Band structures (a,d,g) and trivial (b,e,h) and topological (c,f,i) loops of pSSH (a,b,c), aSSH (d,e,f) and daSSH (g,h,i) models. Winding numbers w around the torus in the Brillouin zone are given for the selected bands.

Figure 2. Bulk-heavy boundary correspondence in phonon Su-Schrieffer-Heeger analogue models. Eigenmode displacements (eigenmode no. 27) before and after imposing a heavy boundary condition for a phonon SSH analogue system (pSSH) of 52 units. Mass ratio m_H/m_b increased from 1 to 1.1. $\kappa_1/\kappa_2=3$ (1/3) for the topological (trivial) case. The eigenvalue spectra as a function of the heavy mass for the topological (trivial) phase is show at the left (right).

Extended Data Figure 2: Topological boundary states as a result of bulk-boundary correspondence in the aSSH model

Eigenvalue spectra when weak and strong springs are exchanged for a system of 52 pearls described by the aSSH model for $\theta = 0^\circ$ (**a**) and $\theta = 15^\circ$ (**c**). The starting (ending) value of κ_1 , $\kappa_2 = 3, 1$ (1, 3). In both spectra, a double degenerate (eigenmodes 79 and 80) topological boundary mode (TBM) is recognized when $\kappa_1 > \kappa_2$. This TBM splits in two bulk modes (one optical and one acoustical) when $\kappa_1 < \kappa_2$. **b,d**. Eigenmode maps for the eigenmodes 79 and 80 showing exponential localization next to the heavy mass when $\kappa_1/\kappa_2 = 3$ and delocalization into the bulk when $\kappa_1/\kappa_2 = 1/3$. The longitudinal eigenmode displacement in **b** are shown transversal to the actual movement for convenience.

Extended Figure 3: Topological boundary states as a result of bulk-boundary correspondence in the daSSH model

a. Eigenvalue spectra when weak and strong springs are exchanged for a system of 52 pearls described by the daSSH model. The starting (ending) value of κ_1 , $\kappa_2 = 3, 1$ (1, 3). When $\kappa_1 > \kappa_2$ a double degenerate (eigenmodes 131 and 132) topological boundary mode (TBM) is recognized. For $\kappa_1 < \kappa_2$, the TBM splits in two bulk modes, one acoustical (131 in blue) and one optical (132 in red). **b.** Eigenmode maps for the eigenmodes 131 and 132 showing exponential localization next to the heavy mass when $\kappa_1/\kappa_2 = 3$. When $\kappa_1/\kappa_2 = 1/3$, the acoustical eigenmode 131 is delocalized into the bulk, while the optical one into a high energy out-of-phase eigenmode.

Further, the authors discuss a 2D-like SSH model (Fig. 5). When coupling 1D topological phases, one has to be very careful, as the classification might change significantly. Take e.g. the 2D SSH model, which has very different topological phases than the 1D model. For example, it cannot really be defined using a winding number. Further, the most famous variant of the 2D SSH model, the BHZ model, finds a so-called fragile topology, which is characterized by corner states instead of 1D boundary modes. See e.g. *Science* 357,61–66 (2017) and *PRB* 107, 045118 (2023). What that means is that the authors need to be more precise to help the reader understand which type of topology they are talking about. Right now, this is unclear to me. This will help to substantiate the findings of the paper and will increase its possible impact.

We agree that most topological phases cannot be coupled to higher dimensions. Instead, our goal was to convey that our array can be approximated as weakly-coupled, closed-packed 1D chains. This is substantiated by the patterning of one boundary mode in the array similarly to the double chain (cf. **Figure 6k** vs. **Figure 5c**). This argument is strengthened for supramolecular arrays of 1D hydrogen bonded or metal-organic supramolecular chains. Note also that in the current work we chose a very weakly-coupled array because it conveys the stronger message that weak as well as strongly interacting supramolecular arrays with alternating spring-like strengths may present topological phonon properties.

To raise awareness about 1D vs 2D phases, we have added in the text the suggested caveat “*When coupling 1D topological phases, meticulous attention is necessary, as the classification might change significantly between 1D and 2D phases. Interestingly, the observed patterning of topological boundary states suggests that closely-packed chains of weakly-interacting molecules behave indeed as pseudo-1D double chains*”.

Reviewer #2 (Remarks to the Author):

The authors studied the topological phonon boundary modes in polymer chains and supramolecular lattices on surfaces.

Due to the following reasons, I can not accept this work.

*1. Major thing: The authors stated that 'The study of topology in the context of electronic phases of matter is widespread. Yet vibrational and dynamical topological phases of the matter remain to be explored'. I do not agree this comment. Please note that the topological phonon phases have been widely investigated in solids and artificial crystals, see *Physical Review Letters*, 2016, 117(6): 068001, *Advanced Functional Materials*, 2020, 30(8): 1904784., *Proceedings of the National Academy of Sciences*, 2016, 113(33): E4767-E4775., *Science Advances*, 2020, 6(46): eabd1618., *Physical RevLetters*, 2023, 131(11): 116602, etc... I can list more than 50 papers focusing on topological phonons. The vibrational and dynamical topological phases of matter are not new, and the study is also widespread. Therefore, this is the main reason I must reject this work, I think communications physics is a good home for the current paper.*

We thank the reviewer for the comment. To avoid misinterpretation, we have rephrased the sentence "Yet vibrational and dynamical topological phases of matter remain to be explored" by "Yet dynamic, vibrational topological phases of atomistic soft matter remain to be explored at finite temperature" to state that to the best of our knowledge our work is also the first study to study topological boundary modes and their boundaries under dynamic conditions.

Moreover, as stated in the abstract "These topological phonon boundary modes remain to be studied both theoretically and experimentally in synthetic materials such as polymers and supramolecular assemblies at the atomistic level under thermal fluctuations" our work is the first study of topological phonon properties in soft-matter at interfaces. Additionally, we have now substantiated key functional aspects, such as to the best of our knowledge the first patterning of modes, as elaborated below.

Finally, we regret to have missed 50 important references in the topic. We now include ~40 new references on topological phonons (see the end of the document).

Note that compared with the model for polymer chains and supramolecular lattices, the topological phonons study of 3D realistic materials may be more useful for the follow-up experimental studies. Hence, again, I do not suggest this work be published in NC now; it may be ok three years ago, not now.

Exploring topological phonons in 3D materials is intriguing for a wide variety of applications and condensed matter device physicists, particularly those focused on inorganic materials.

Yet engineering phonons in soft-matter is arresting for many more applications and molecular surface science. As molecular surface scientists and supramolecular chemists, our realistic 1D and 2D materials appeal to a larger material chemistry and interdisciplinary community. There are arguments for the combinations of chemically stable organic monolayers being a larger set than inorganic materials. To this end we introduce two new SSH-like phonon models which can be engineered at surfaces with organic monolayers, as opposed to searching databases for crystalline phonons. Such possibility of engineering monolayers and their boundary modes could lead to patterning topological phonon circuitry. To provide evidence of the above, we now, for the first time in the literature to the best of our knowledge, pattern topological boundary modes in realistic atomistic materials by increasing

the mass of a unit, something we believe could lead to arresting studies in the field, cf. Figure below. Note how the boundary mode appears mostly at the right-hand-side of the patterning.

(Figure 6) i. Supramolecular ribbon with heavier inner molecules, marked in red, whose mass has been increased three-fold. j. NMA of the system showing an inner boundary eigenmode at 20 cm^{-1} and whose mapping is shown in (k). k. Eigenmode mapping at 20 cm^{-1} , mostly localized at the molecules with a stronger interaction next (r.h.s.) to the heavy ones as depicted in (i).

To summarize, the study and classification of topological phonons in soft-matter materials is a new endeavour worth pursuing. We also consider that *Nature Communications* is the right interdisciplinary venue to publish novel application to new systems, like with this one for example <https://www.nature.com/articles/ncomms2451>

2. Minor thing: The authors mentioned many times about the topological boundary mode. However, the authors did not prove the boundary mode is topologically nontrivial. At least you can understand the boundary based on the nontrivial topological index to show the readers that the boundary is topologically nontrivial.

We have now characterized the trivial and topological phase by a winding number, 0 and ± 1 respectively, for the three models and expanded the **Supplementary Figure 1** accordingly.

Supplementary Figure 1. Band structures (a,d,g) and geometric phases for the trivial (b,e,h) and topological (c,f,i) phases of pSSH (a,b,c), aSSH (d,e,f) and daSSH (g,h,i) models. Winding numbers w around the torus in the Brillouin zone are given for the selected bands.

We have also introduced the bulk-heavy boundary correspondence in a new section to unambiguously demonstrate the topological character of the boundary mode, showing that only for the topological index of $w=1$ a boundary mode is expressed.

Reviewer #3 (Remarks to the Author):

The manuscript describes a series of Su-Schrieffer-Heeger (SSH) phonon models and their topological features. Specifically, the models discussed are the traditional chain, a chain adsorbed on a substrate, a double chain adsorbed on a substrate, and a supramolecular lattice. These models are then associated with real polymers and derived materials, and their finite temperature properties simulated using molecular dynamics. The various topological edge modes are characterized, and a specific prediction is their propagation upon the excitation of single molecules.

While topological phonons is an interesting field of research, it is unclear that Nature Communications is the best journal for this work. There is a significant body of work on topological phonons (which the authors seem to be unaware of), and it is unclear that the results presented here provide any fundamentally new insights or discover novel phenomena associated with topological phonons. In particular:

1. The authors write: "Thus far, mostly mechanical models, such as Maxwell lattices, have been employed to study vibrational topological properties". In fact, there is a very large body of literature exploring topological phonons in atomistic systems both theoretically and experimentally, with a non-exhaustive list including:

* One of the early experimental observations of topological phonons: https://scholar.google.com/citations?view_op=view_citation&hl=zh-CN&user=thgksIEAAAJ&citation_for_view=thgksIEAAAJ:zYLM7Y9cAGgC

* Experimental observation of topological phonons in graphene: <https://journals.aps.org/prl/abstract/10.1103/PhysRevLett.131.116602>

* And a recent full catalogue of topological phonons in thousands of materials: <https://arxiv.org/abs/2211.11776>

* Database associated with the above work: <https://www.topologicalquantumchemistry.fr/topophonons/>

In this context, the present work provides yet another characterization of topological phonons in atomistic systems. It is unclear whether there is something special about the systems studied in this work compared to many earlier works (and I emphasize again that the list above is by no means exhaustive).

We thank the reviewer and regret that we did not properly introduce the body of literature in our work.

We have now introduced ~40 new references regarding topological phonons (see below).

We have adapted the introduction to clearly, in a step wise manner convey that:

–The body of work on topological phonons is exhaustive, but we study them in atomistic soft-matter for the first time:

"The study of topology in the context of electronic phases of matter is widespread^{1–6}. Equivalent topological concepts have been also applied to phononic system^{7–16}, in both crystalline materials^{17–24} and periodic artificial structures^{25–32}. Yet dynamic³³, vibrational³⁴ topological phases of atomistic soft matter remain to be explored at finite temperatures: chemical systems such as polymers, self-assembled networks, metal-organic frameworks, and covalent organic frameworks, to mention a few."

–We study the dynamics of the topological phonons under thermal fluctuations:

"Second, we introduce the adsorbed SSH (aSSH) model on an implicit surface and depict its topological nontrivial boundary modes in atomistic simulations under thermal fluctuations."

–Phonons in topological metamaterials grant access to the engineering of boundary modes, and, in the same spirit, organic layers grant access to the engineering of boundary modes. We substantiate this by patterning heavy-boundary modes in soft-matter for the first time:

“Finally, we deal with the extension of the daSSH model to identify and pattern topological boundary modes in an all-atomistic supramolecular ribbon, and study the effect of topological flat band excitation and propagation by comparison to a free boundary mode.”
To summarize:

“Our results reveal for the first time in the literature, topological boundary patterning principle and bulk-heavy boundary correspondence, phonon topology under thermalized conditions, and introduces new material platforms for investigating topological physics at the interface between organic chemistry and condensed matter physics.”

2. *On the methodological front, can the authors please expand on the meaning of MD simulations at 1K or 10K? Wouldn't the dynamics of the system at that temperature be dominated by quantum nuclear effects, which as far as I understand are not included in the calculations presented?*

Vibrational quantum effects are indeed important for the study of low-temperature phonon dynamics. However, the low energy modes of large molecules are expected to arise from average movement of many degrees of freedom wherein the role of vibrational quantum effects could be mitigated. To substantiate this, we have now written in the methods section that force fields reproduce the dynamics of large molecules with very high accuracy below 80 K, e.g. rotational rates have been previously shown to agree within 9 meV to experiment (<https://pubs.acs.org/doi/10.1021/nl5014162>) and can reproduce probability densities (<https://www.nature.com/articles/ncomms7210>).

3. *Can the authors expand on the propagation vs. non-propagation nature of boundary modes? Is this a completely unambiguous marker of topological phonons?*

Differences in propagation of boundary modes are not an unambiguous marker for topology, but we claim are a plausible consequence of topological phonons. We now write in the main text *“Differences in propagation between trivial and topological boundary modes are not an unambiguous marker for topology, but are a plausible consequence of topological phonons stemming from topological robustness against perturbation. Trivial and simply localized boundary phonon modes are disrupted by the phonon excitation, unlike topological boundary modes whose eigendisplacements are exponentially localized.”*

4. *More generally, could the authors clarify what is special about topological phonons in these particular materials that merits the report of this work in Nature Communications? Are there new fundamental insights into topological phonons? Or are there particular applications that this class of material would enable but others wouldn't? Something else?*

We appreciate the reviewer's comment which motivated underscoring why are supramolecular systems an exciting platform for the study of topological phonons towards applications.

We have now highlighted in the main text few insights, which justify the topological phonons in organic supramolecular systems at interfaces as a unique field of its own, such as the heavy boundary-bulk correspondence, the robustness under thermal fluctuations, patterning and the dynamics of their boundary modes. We now provide analytical expressions for the

bulk-heavy boundary correspondence and interpretation of the robustness towards excitation which we consider fundamental insights.

Moreover, we demonstrate with a new Figure (below), that the bulk-heavy boundary correspondence allows the patterning of topological boundary modes of diverse shapes in supramolecular lattices. While this strategy could be adopted for 2D materials by atomic replacement with heavier elements, in supramolecular systems, a single hydrogen adsorbed on a molecule or the combination of two different molecular species (one heavier than the other) should provoke the manifestation of topological boundary modes, a consequence of the bulk-heavy boundary correspondence we now elaborate in Figure 2.

(Figure 6) **i.** Supramolecular ribbon with heavier inner molecules, marked in red, whose mass has been increased three-fold. **j.** NMA of the system showing an inner boundary eigenmode at 20 cm^{-1} and whose mapping is shown in **(k)**. **k.** Eigenmode mapping at 20 cm^{-1} , mostly localized at the molecules with a stronger interaction next (r.h.s.) to the heavy ones as depicted in (i).

New references on topological acoustics and phononics:

1. Zhang, L., Ren, J., Wang, J.-S. & Li, B. Topological Nature of the Phonon Hall Effect. *Phys. Rev. Lett.* **105**, 225901 (2010).
2. Sun, K., Souslov, A., Mao, X. & Lubensky, T. C. Surface phonons, elastic response, and conformal invariance in twisted kagome lattices. *PNAS* **109**, 12369–12374 (2012).
3. Zhang, L. & Niu, Q. Chiral Phonons at High-Symmetry Points in Monolayer Hexagonal Lattices. *Phys. Rev. Lett.* **115**, 115502 (2015).
4. Stenull, O., Kane, C. L. & Lubensky, T. C. Topological Phonons and Weyl Lines in Three Dimensions. *Phys. Rev. Lett.* **117**, 068001 (2016).
5. Po, H. C., Bahri, Y. & Vishwanath, A. Phonon analog of topological nodal semimetals. *Phys. Rev. B* **93**, 205158 (2016).
6. Liu, Y., Lian, C.-S., Li, Y., Xu, Y. & Duan, W. Pseudospins and Topological Effects of Phonons in a Kekulé Lattice. *Phys. Rev. Lett.* **119**, 255901 (2017).
7. Liu, Y., Xu, Y. & Duan, W. Berry phase and topological effects of phonons. *National Science Review* **5**, 314–316 (2018).
8. Po, H. C., Vishwanath, A. & Watanabe, H. Symmetry-based indicators of band topology in the 230 space groups. *Nat Commun* **8**, 50 (2017).
9. Bradlyn, B. *et al.* Topological quantum chemistry. *Nature* **547**, 298–305 (2017).
10. Liu, Y., Xu, Y., Zhang, S.-C. & Duan, W. Model for topological phononics and phonon diode. *Phys. Rev. B* **96**, 064106 (2017).
11. Fang, C., Gilbert, M. J., Dai, X. & Bernevig, B. A. Multi-Weyl Topological Semimetals Stabilized by Point Group Symmetry. *Phys. Rev. Lett.* **108**, 266802 (2012).
12. Weng, H., Fang, C., Fang, Z. & Dai, X. Coexistence of Weyl fermion and massless triply degenerate nodal points. *Phys. Rev. B* **94**, 165201 (2016).
13. Zhang, T. *et al.* Double-Weyl Phonons in Transition-Metal Monosilicides. *Phys. Rev. Lett.* **120**, 016401 (2018).
14. Li, J. *et al.* Coexistent three-component and two-component Weyl phonons in TiS, ZrSe, and HfTe. *Phys. Rev. B* **97**, 054305 (2018).
15. Zhu, H. *et al.* Observation of chiral phonons. *Science* **359**, 579–582 (2018).
16. Jin, Y., Wang, R. & Xu, H. Recipe for Dirac Phonon States with a Quantized Valley Berry Phase in Two-Dimensional Hexagonal Lattices. *Nano Lett.* **18**, 7755–7760 (2018).
17. Peng, B., Hu, Y., Murakami, S., Zhang, T. & Monserrat, B. Topological phonons in oxide perovskites controlled by light. *Science Advances* **6**, eabd1618 (2020).
18. Wang, X. *et al.* Topological nodal line phonons: Recent advances in materials realization. *Applied Physics Reviews* **9**, 041304 (2022).
19. Yang, Z. *et al.* Topological Acoustics. *Phys. Rev. Lett.* **114**, 114301 (2015).
20. Xiao, M. *et al.* Geometric phase and band inversion in periodic acoustic systems. *Nature Phys* **11**, 240–244 (2015).
21. Khanikaev, A. B., Fleury, R., Mousavi, S. H. & Alù, A. Topologically robust sound propagation in an angular-momentum-biased graphene-like resonator lattice. *Nat Commun* **6**, 8260 (2015).
22. He, H. *et al.* Topological negative refraction of surface acoustic waves in a Weyl phononic crystal. *Nature* **560**, 61–64 (2018).
23. Serra-Garcia, M. *et al.* Observation of a phononic quadrupole topological insulator. *Nature* **555**, 342–345 (2018).
24. Wu, X. *et al.* Topological phononics arising from fluid-solid interactions. *Nat Commun* **13**, 6120 (2022).
25. Xue, H., Yang, Y. & Zhang, B. Topological acoustics. *Nat Rev Mater* **7**, 974–990 (2022).
26. Xu, Y. *et al.* Catalogue of topological phonon materials. Preprint at <https://doi.org/10.48550/arXiv.2211.11776> (2022).
27. Petretto, G. *et al.* High-throughput density-functional perturbation theory phonons for inorganic materials. *Sci Data* **5**, 180065 (2018).

28. Li, J. *et al.* Computation and data driven discovery of topological phononic materials. *Nat Commun* 12, 1204 (2021).
29. Süsstrunk, R. & Huber, S. D. Observation of phononic helical edge states in a mechanical topological insulator. *Science* 349, 47–50 (2015).
30. Peano, V., Brendel, C., Schmidt, M. & Marquardt, F. Topological Phases of Sound and Light. *Phys. Rev. X* 5, 031011 (2015).
31. Li, X. *et al.* Su-Schrieffer-Heeger model inspired acoustic interface states and edge states. *Applied Physics Letters* 113, 203501 (2018).
32. Maldovan, M. Sound and heat revolutions in phononics. *Nature* 503, 209–217 (2013).
33. Jensen, J. S. Phononic band gaps and vibrations in one- and two-dimensional mass-spring structures. *Journal of Sound and Vibration* 266, 1053–1078 (2003).
34. Yin, J. *et al.* Band transition and topological interface modes in 1D elastic phononic crystals. *Sci Rep* 8, 6806 (2018).
35. Xia, B., Liu, H. & Liu, F. Negative interatomic spring constant manifested by topological phonon flat band. *Phys. Rev. B* 109, 054102 (2024).
36. Kariyado, T. & Hatsugai, Y. Hannay Angle: Yet Another Symmetry-Protected Topological Order Parameter in Classical Mechanics. *J. Phys. Soc. Jpn.* 85, 043001 (2016)
37. Xiao, M. *et al.* Geometric phase and band inversion in periodic acoustic systems. *Nature Phys* 11, 240–244 (2015)

New references on experimental observation of phononic modes in graphene and metal silicides:

1. Li, J. *et al.* Direct Observation of Topological Phonons in Graphene. *Phys. Rev. Lett.* 131, 116602 (2023).
2. Miao, H. *et al.* Observation of Double Weyl Phonons in Parity-Breaking FeSi. *Phys. Rev. Lett.* 121, 035302 (2018).
3. Jin, Z. *et al.* Chern numbers of topological phonon band crossing determined with inelastic neutron scattering. *Phys. Rev. B* 106, 224304 (2022).